# Structure of the connexin-43 gap junction channel in a putative closed state

Chao Qi[1,2†], Silvia Acosta Gutierrez[3,4†], Pia Lavriha[1,2], Alaa Othman[5], Diego Lopez-Pigozzi[6,7], Erva Bayraktar[7], Dina Schuster[1,2,5], Paola Picotti[5], Nicola Zamboni[5], Mario Bortolozzi[6,7], Francesco Luigi Gervasio[8,9,10]*, Volodymyr M Korkhov[1,2]*

[1]Institute of Molecular Biology and Biophysics, ETH Zurich, Zurich, Switzerland; [2]Laboratory of Biomolecular Research, Paul Scherrer Institute, Villigen, Switzerland; [3]Institute for the Physics of Living Systems, Institute of Structural and Molecular Biology, University College London, London, United Kingdom; [4]Institute for Bioengineering of Catalunya (IBEC), The Barcelona Institute of Science and Technology, Barcelona, Spain; [5]Institute of Molecular Systems Biology, ETH Zurich, Zurich, Switzerland; [6]Department of Physics and Astronomy "G. Galilei", University of Padova, Padua, Italy; [7]Veneto Institute of Molecular Medicine (VIMM), Padua, Italy; [8]Department of Chemistry, University College London, London, United Kingdom; [9]School of Pharmaceutical Sciences, University of Geneva, Geneva, Switzerland; [10]ISPSO, University of Geneva, Geneva, Switzerland

*For correspondence:
francesco.gervasio@unige.ch
(FLG);
volodymyr.korkhov@psi.ch (VMK)

†These authors contributed
equally to this work

Competing interest: The authors
declare that no competing
interests exist.

Reviewing Editor: David Drew,
Stockholm University, Sweden

**Abstract** Gap junction channels (GJCs) mediate intercellular communication by connecting two neighbouring cells and enabling direct exchange of ions and small molecules. Cell coupling via connexin-43 (Cx43) GJCs is important in a wide range of cellular processes in health and disease (Churko and Laird, 2013; Liang et al., 2020; Poelzing and Rosenbaum, 2004), yet the structural basis of Cx43 function and regulation has not been determined until now. Here, we describe the structure of a human Cx43 GJC solved by cryo-EM and single particle analysis at 2.26 Å resolution. The pore region of Cx43 GJC features several lipid-like densities per Cx43 monomer, located close to a putative lateral access site at the monomer boundary. We found a previously undescribed conformation on the cytosolic side of the pore, formed by the N-terminal domain and the transmembrane helix 2 of Cx43 and stabilized by a small molecule. Structures of the Cx43 GJC and hemichannels (HCs) in nanodiscs reveal a similar gate arrangement. The features of the Cx43 GJC and HC cryo-EM maps and the channel properties revealed by molecular dynamics simulations suggest that the captured states of Cx43 are consistent with a closed state.

## eLife assessment

Gap junctions, formed from connexins, are important in cell communication, allowing ions and small molecules to move directly between cells. By determining the Cryo EM structure of the structure of connexin 43 in a putative closed state involving lipids, the study makes an **important** contribution to the development of a mechanistic model for connexin activation. The connexin 43 structure is **solid** and its presentation will appeal to the channel and membrane protein communities.

## Introduction

Gap junction (GJ)-mediated intercellular communication is one of the major pathways of information exchange between the cells. GJs are specialized regions of the plasma membrane at the cell–cell

interface that link two adjacent cells and establish their metabolic and electrical coupling (*Rodríguez-Sinovas et al., 2021*). Connexins are the building blocks of the GJ channels (GJCs) which belong to a group of large-pore channels. This group includes a number of structurally related (innexins, pannexins, and LRRC8) and unrelated proteins (CALHM) (*Syrjanen et al., 2021*). A total of 21 connexin genes have been identified in the human genome (*Srinivas et al., 2018*). The 43 kDa connexin-43 (Cx43, gene name *GJA1*) was identified as a major constituent of rat heart GJs in 1987 (*Beyer et al., 1987*), and it is arguably one of the most extensively studied connexins. Like all connexin homologues, Cx43 monomers assemble into hexameric hemichannels (HCs), also known as connexons. HCs that reach the plasma membrane of one cell may interact with their counterparts on the neighbouring cell, forming GJCs, typically organized into hexagonal arrays at the intercellular interface (*Robertson, 1963*). GJCs enable direct metabolic and electric coupling between the cells, facilitating the passage of ions, small molecules, metabolites, peptides, and other cellular components below a size threshold of approximately 1.5 kDa (*Laird and Lampe, 2018*). GJCs formed by Cx43 are crucial for a wide range of physiological processes, from propagation of heart action potentials (*Poelzing and Rosenbaum, 2004*) to maintenance of neuro-glial syncytium (*Liang et al., 2020*) and skin wound healing (*Churko and Laird, 2013*). The clinical importance of Cx43 is highlighted by mutations linked to several genetic disorders, such as oculodentodigital dysplasia (ODDD) (*Jamsheer et al., 2014Paznekas et al., 2003*; *Jamsheer et al., 2014*; *Paznekas et al., 2009*; *Wiest et al., 2006*), hypoplastic left heart syndrome 1 (*Dasgupta et al., 2001*), Hallermann–Streiff syndrome (*Pizzuti et al., 2004*), and atrioventricular septal defect 3 (*Dasgupta et al., 2001*), and by recognition of the protein as a drug target for treatment of cancer, skin wounds, eye injury and inflammation, and cardiac arrhythmias (reviewed by *Laird and Lampe, 2018*).

Much of what we know today about the molecular biology, electrophysiological properties, and regulation of connexin gap junctions has been derived from the studies on Cx43 (reviewed in *Ribeiro-Rodrigues et al., 2017*). Early attempts to characterize the structure of Cx43 using cryo-electron microscopy (cryo-EM) of 2D crystals produced low resolution reconstructions (*Unger et al., 1999*). More recently, several homologous connexin GJCs and HCs have been structurally characterized at high resolution, using X-ray crystallography and cryo-EM. The structures of Cx26 (*Maeda et al., 2009*) and Cx46/50 GJCs (*Myers et al., 2018*; *Flores et al., 2020*), together with the recent structure of Cx31.3 HC (*Lee et al., 2020*) provided deep insights into the shared structural features of the connexin channels. These structures hint at the role of the N-terminal domain (NTD) in molecular gating of GJCs and HCs, with several conformations of the NTD observed in different connexin homologues (*Myers et al., 2018*; *Flores et al., 2020*; *Lee et al., 2020*). Additionally, biochemical evidence points to the roles in channel gating played by the link between Cx43 intracellular loop and the C-terminal region (*Xu et al., 2012*; *Ponsaerts et al., 2010*). However, despite the availability of this evidence, the molecular determinants of intracellular connexin channel gating remain unclear. To shed light on the structural basis of Cx43 gating, we set out to determine its structure by cryo-EM and to analyse its dynamics with molecular dynamics (MD) simulations.

## Results

### Structures of Cx43 GJC and HC in detergent micelles and in nanodiscs

Our Cx43 expression system was tested using electrophysiology in HeLa cells (*Figure 1—figure supplement 1A–D*). The experiments confirmed the functionality of Cx43 GJCs expressed in transfected cells. For large-scale protein expression, Cx43 featuring a C-terminal HRV3C-YFP-twinStrep tag, was expressed in adherent mammalian cells (HEK293F) using transient transfection method. The produced protein was purified using affinity and size exclusion chromatography in digitonin (*Figure 1—figure supplement 2A–C*) and flash frozen on cryo-EM grids. The Coomassie-stained sodium dodecyl sulfate–polyacrylamide gel electrophoresis (SDS–PAGE) gel bands of the purified protein (*Figure 1—figure supplement 2C*) are consistent with the western blot analysis (*Figure 1—figure supplement 2D*) of the expressed untagged Cx43. The grids were subjected to single particle cryo-EM analysis (*Figure 1—figure supplement 3*), yielding the final 3D reconstruction at 2.26 Å resolution (*Figure 1A*, *Figure 1—figure supplements 3 and 4*, *Table 1*).

The overall architecture of the protein complex resembles that observed with other connexin GJCs, with two HCs (connexons) from adjacent plasma membrane regions coupled to form a full

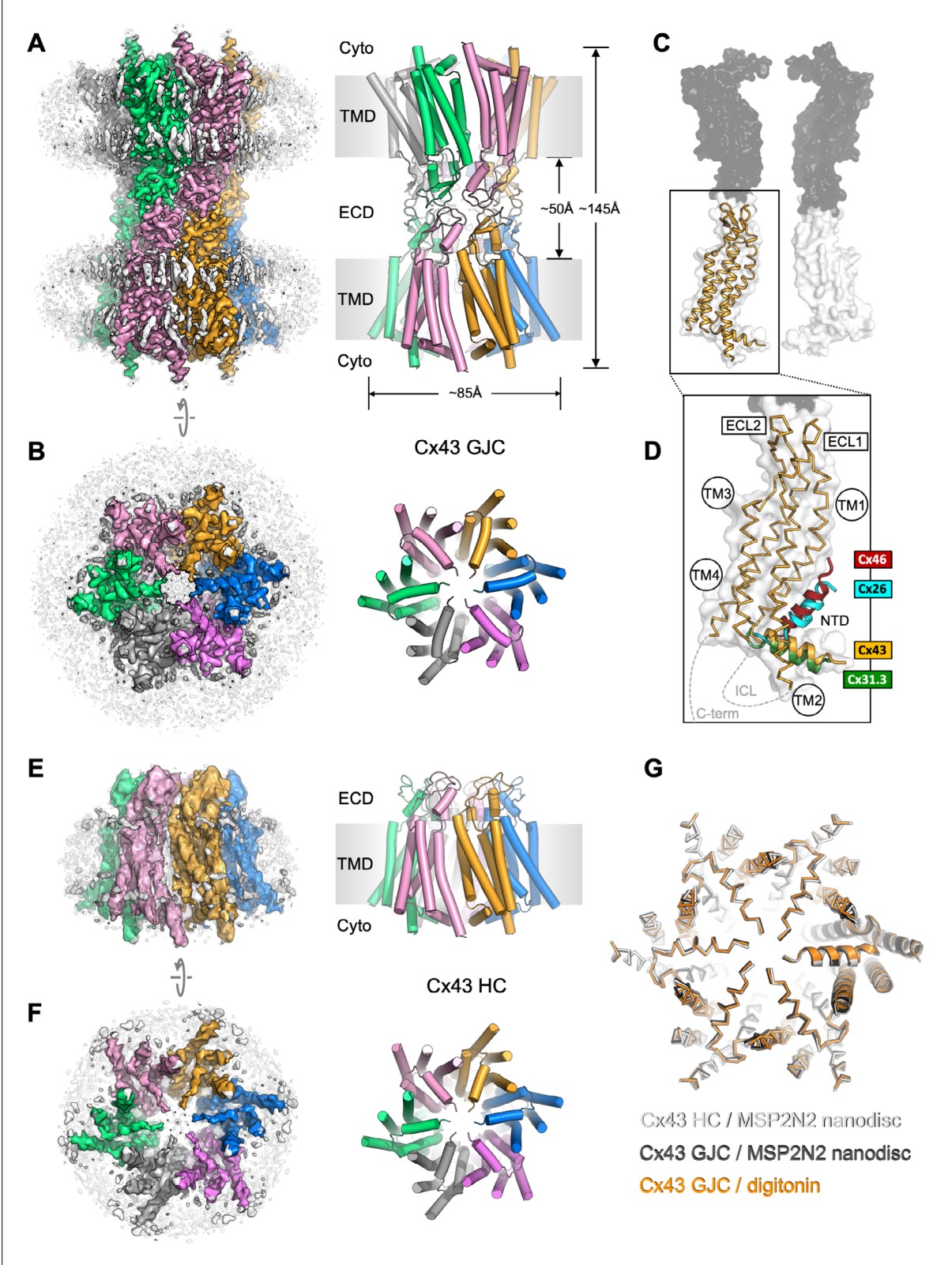

**Figure 1.** Structure of human connexin-43 (Cx43) gap junction channel (GJC). (**A, B**) Cryo-EM density map and model of Cx43 GJC solved by cryo-EM at 2.26 Å resolution. The individual Cx43 monomers in each hemichannel (HC) within the GJC are coloured blue, pink, grey, green, salmon, and orange. Grey densities correspond to the detergent micelle and the bound sterol-like molecules. (**C**) The position of a single Cx43 monomer (orange) within a GJC (represented as a surface of juxtaposed Cx43 monomers in two distinct membrane regions, white and grey). (**D**) Alignment of the monomers

*Figure 1 continued on next page*

*Figure 1 continued*

of Cx43, Cx26 (PDB ID: 2zw3), Cx46 (PDB ID: 7jkc), and Cx31.3 (PDB ID: 6l3t) shows the distinct orientations of the N-terminal domain (NTD) helix. Individual TM domains, extracellular loops 1 and 2 (ECL1–2), relative positions of the intracellular loop (ICL), and the C-terminus (C-term) are indicated. (**E, F**) Same as A, B, for the Cx43 HC in MSP2N2 nanodiscs at 3.98 Å resolution. (**G**) Alignment of three indicated structures shows that the conformation of the gating region is highly conserved. The models are shown as ribbons; one of the aligned monomers in each of the structures is represented as cartoon, to clearly indicate the monomer boundaries.

The online version of this article includes the following source data and figure supplement(s) for figure 1:

**Figure supplement 1.** Localization and function of connexin-43 (Cx43) gap junction channels (GJCs).

**Figure supplement 2.** Expression and purification of connexin-43 (Cx43).

**Figure supplement 2—source data 1.** The data corresponding to the panel shown in *Figure 1—figure supplement 2B* (size exclusion chromatography raw data).

**Figure supplement 2—source data 2.** The data corresponding to the panel shown in *Figure 1—figure supplement 2C*.

**Figure supplement 2—source data 3.** The data corresponding to the panel shown in *Figure 1—figure supplement 2C*.

**Figure supplement 2—source data 4.** The data corresponding to the panel shown in *Figure 1—figure supplement 2D*.

**Figure supplement 2—source data 5.** The data corresponding to the panel shown in *Figure 1—figure supplement 2D*.

**Figure supplement 2—source data 6.** The data corresponding to the panel shown in *Figure 1—figure supplement 2E* (size exclusion chromatography raw data).

**Figure supplement 2—source data 7.** The data corresponding to the panel shown in *Figure 1—figure supplement 2F*.

**Figure supplement 2—source data 8.** The data corresponding to the panel shown in *Figure 1—figure supplement 2F*.

**Figure supplement 3.** Cryo-EM image processing workflow of connexin-43 (Cx43) in digitonin.

**Figure supplement 4.** Fourier shell correlation (FSC) of the connexin-43 (Cx43) reconstruction, local resolution and density map features.

**Figure supplement 5.** Mass spectrometric characterization of purified connexin-43 (Cx43).

**Figure supplement 6.** Cryo-EM image processing workflow for connexin-43 (Cx43) gap junction channel (GJC) in nanodisc.

**Figure supplement 7.** Cryo-EM image processing procedure for connexin-43 (Cx43) hemichannel (HC) in nanodisc.

**Figure supplement 8.** Fourier shell correlation (FSC) and local resolution maps of connexin-43 (Cx43) in nanodiscs.

**Figure supplement 9.** Comparison of the three cryo-EM structures of connexin-43 (Cx43) and sequence alignment of N-terminal domain (NTD) and TM2.

GJC (*Figure 1A, B*). The exterior of the channel is decorated by several ordered detergent and/or lipid-like density elements (*Figure 1A, B*), reminiscent of the previously observed lipids bound at the protein–bilayer interface in other connexins (*Flores et al., 2020*). In the case of Cx43, the lipid-like densities appear to decorate the protein–lipid bilayer interface at both the inner and the outer leaflet of the membrane. Comparison of the Cx43 monomers to the available structures of Cx26 and Cx46 GJCs and the Cx31.3 HC revealed a major difference in the NTD arrangement in these channels (*Figure 1C, D*). The conformation of the NTD in the Cx31.3 HC structure appears to be the closest to that in Cx43 GJC. Analysis of the tryptic peptides revealed that most of the protein lacks the residue M1 (*Figure 1—figure supplement 5*), and thus the model was built starting with G2.

To ascertain that the conformation of Cx43 GJC is not induced by the detergent present in the sample, for example through detergent binding at specific sites on the protein surface, we removed the detergent and reconstituted the protein into MSP2N2 nanodiscs, using 1-palmitoyl-2-oleoyl-glycero-3-phosphocholine (POPC) as a mimic for the native lipid environment. The reconstituted protein was subjected to the same imaging and analysis workflow as Cx43 GJC in digitonin. The 2D classes of the MSP2N2-reconstituted Cx43 showed features consistent with a mixture of GJCs and HC. Processing of the corresponding particles resulted in 3D reconstructions of the Cx43 GJC at 2.95 Å resolution (*Figure 1—figure supplements 6 and 8*) and HC in nanodiscs at 3.98 Å resolution (*Figure 1E, F*, *Figure 1—figure supplements 7 and 8*). The GJCs in detergent and in nanodiscs are nearly identical, with a root-mean-square deviation (RMSD) of 0.97 Å between the aligned Cx43 monomers (*Figure 1G*, *Figure 1—figure supplement 9A*). Although the differences between the HC in nanodisc and the GJC are more pronounced (*Figure 1—figure supplement 9A*), the conformation of the cytosolic region and the NTD is highly conserved (*Figure 1G*).

**Table 1.** Cryo-EM data collection and image processing statistics.

| Data collection | | | |
| --- | --- | --- | --- |
| Sample | Cx43 (Digitonin) | Cx43_nanodisc | |
| Instrument | FEI Titan Krios/Gatan K3 Summit/ Quantum GIF | FEI Titan Krios/Gatan K3 Summit/ Quantum GIF | |
| | | Cx43 GJC | Cx43 HC |
| Voltage | 300 | | |
| Electron dose (e⁻/Å) | 50 | | |
| Defocus range (μm) | −1 to −2 | | |
| Pixel size (Å) | 0.654 | | |
| Map resolution (Å) FSC threshold 0.143 | 2.26 | 2.95 | 3.98 |
| Map sharping *b*-factor (Å) | −49 | −86 | −100 |
| Number of particles | 50,471 | 10,886 | 20,526 |
| **Refinement** | | | |
| Model resolution (Å) FSC threshold 0.5 | 2.3 | 3.1 | 4.2 |
| Map CC | 0.84 | 0.84 | 0.77 |
| Model composition | | | |
| Protein residues | 2268 | 2268 | 1134 |
| ADP (*B* factor) | 31.90 | 40.02 | 158.95 |
| Bond length RMSD (Å) | 0.003 | 0.003 | 0.003 |
| Bond angle RMSD (°) | 0.552 | 0.514 | 0.678 |
| Validation | | | |
| MolProbity score | 1.47 | 1.37 | 1.79 |
| Clash score | 8.80 | 6.76 | 19.75 |
| Rotamer outliers (%) | 0 | 0 | 0 |
| **Ramachandran plot** | | | |
| Favoured (%) | 98.92 | 98.24 | 98.92 |
| Allowed (%) | 1.08 | 1.76 | 1.08 |
| Disallowed (%) | 0 | 0 | 0 |

## Conformation of the Cx43 putative gate region

The reconstructed putative gate region of Cx43 shows several narrow openings connecting the pore vestibule and the pore interior: a single ~6–7 Å wide central opening and six adjacent openings of similar dimensions (*Figure 2A, B*). This arrangement of the gate region is distinct from any of the previously observed states of connexin GJCs or HCs. This particular gate conformation is established through an interplay of two structural elements: the NTD and the TM2 (*Figure 2C, D*). The NTD of Cx43 is arranged near parallel to the membrane plane, in a centre-oriented conformation (*Figure 2C*). The HC of Cx31.3 assembles in a similar manner, with a well-ordered NTD resolved by cryo-EM (*Figure 2C, D*). However, the TM2 region of Cx31.3 forms a tight seal with the NTD (*Figure 2E*). In contrast, the TM2 of Cx43 is shifted away from the pore centre, creating six openings (*Figure 2F*). Alignment of the protomers of Cx43 and Cx31.3 showed an RMSD of 1.78 Å (using cealign in PyMol). In contrast, the RMSD value for the region corresponding to the NTD and TM2 was 5.675 Å (calculated using rms_cur in PyMol, with the residue range selections of 2–47 and 2–45 for the aligned protomers of Cx43 and Cx31.3, respectively), further confirming that these two regions differ substantially between Cx43 and

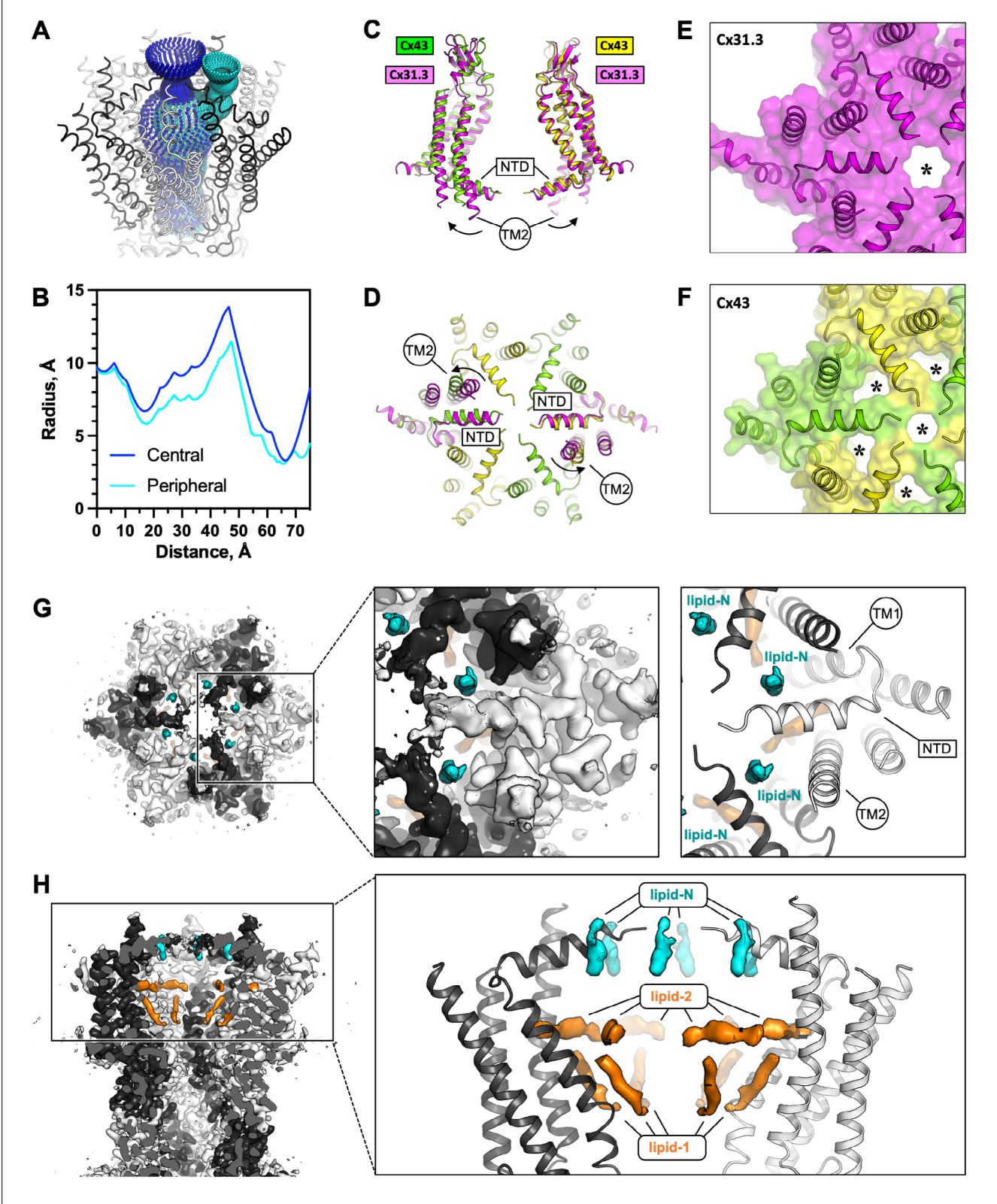

**Figure 2.** The connexin-43 (Cx43) gate adopts a closed conformation. (**A, B**) Analysis of the pore opening dimensions using HOLE reveals a constriction of the pore in the putative gate region. Only a central opening and one of the six peripheral openings within one hemichannel (HC) of Cx43 gap junction channel (GJC) in digitonin is shown. The distance is calculated from the centre of the GJC pore to a point outside of the channel. Central and peripheral openings are coloured blue and cyan, respectively. (**C, D**) Comparison of the Cx43 in digitonin with the Cx31.3 HC shows that the peripheral

*Figure 2 continued on next page*

*Figure 2 continued*

opening is created by the particular arrangement of the N-terminal domain (NTD) and by adjustment of the TM2. The distinct NTD/TM2 arrangement results in a single pore opening in the structure of Cx31.3 (indicated with an asterisk, **E**), contrary to Cx43 (**F**). (**G**) A view of the Cx43 gating region from the cytosol reveals the location of the 'lipid-N' molecules stabilizing the NTD arrangement, shown as isolated densities (cyan). (**H**) A slab view of the gating region parallel to the membrane plane shows the relative arrangement of the lipid-N and the intra-pore lipid densities ('lipid-1' and 'lipid-2'; orange).

The online version of this article includes the following figure supplement(s) for figure 2:

**Figure supplement 1.** Comparison of intrapore lipid densities in detergent and in nanodiscs and lipidomic analysis of connexin-43 (Cx43).

Cx31.3. The arrangement of the Cx43 NTD appears to be stabilized by a small molecule: a density element likely corresponding to a bound small molecule is present within this region (referred to as the 'NTD lipid site'), wedged between the adjacent NTDs (*Figure 2G*). Thus, although the conformation of the Cx43 gate features an opening, this site is blocked by a yet unidentified small molecule (which we refer to as 'lipid-N').

While a direct comparison of the Cx43 gating regions of the Cx43 and Cx31.3 HCs is possible (*Figure 1—figure supplement 9C*), the recently determined 3D reconstruction of the Cx26 mutant N176Y was not accompanied by a new atomic model (*Khan et al., 2021*). Instead the study reporting this reconstruction compared the density map with an HC model from a Cx26 GJC structure (PDB ID: 5ERA) (*Bennett et al., 2016*). As this model lacks the NTD, any comparisons of the gating region in the Cx43 HC with that in the Cx26 HC are presently limited (*Figure 1—figure supplement 9D*).

In addition to the lipid-N molecules stabilizing the NTD arrangement, the cryo-EM density of Cx43 GJC features several well resolved densities inside the pore region (lipid-1 and -2, *Figure 2H*). These elements likely correspond to bound sterol molecules, such as cholesterol co-purified with the protein from the mammalian cells or cholesterol hemisuccinate (CHS) added to the solubilization mixture during protein extraction from the membrane. Similar densities are present in the Cx43 GJC reconstruction in nanodiscs, in the absence of detergent molecules (*Figure 2—figure supplement 1A*). Additionally, it is noteworthy that the annular lipid densities are conserved in both the detergent-solubilized and the nanodisc-reconstitued Cx43 GJCs, indicating that the protein–lipid interface of Cx43 features sites where ordered lipid molecules may bind (*Figure 2—figure supplement 1B*). The functional significance of the ordered annular lipids for the channel activity and for GJ plaque assembly remains to be carefully investigated.

Identification of the lipid-like molecules bound to Cx43 is a significant challenge. Despite the high overall resolution of the 3D reconstruction of the Cx43 GJC (2.26 Å resolution in detergent and 2.95 Å in nanodisc; *Figure 1—figure supplements 3 and 6*), the cryo-EM map features in regions corresponding to the lipid densities are insufficient to assign the lipid identity unambiguously. To determine the identities of these lipids experimentally, we performed lipidomic analysis of the organic extracts prepared from the purified Cx43 samples and compared them to the mock control (extracts of prepared from eluates of the purification procedure using cells that do not overexpress the protein as starting material). The results showed that several types of phospholipids, as well as cholesterol were present in both samples. The only lipid-like compound specifically enriched in the purified sample was dehydroepiandrosterone (DHEA; *Figure 2—figure supplement 1C*). DHEA is a most abundantly expressed neurosteroid known to modulate ligand-gated ion channels, such as GABA$_A$ (*Gartside et al., 2010*) or NMDA and AMPA receptors (*Kimonides et al., 1998*). Although we are presently not able to conclusively state that DHEA corresponds to any of the densities present in our reconstruction, its enrichment in the sample is an interesting observation that deserves detailed future investigations. While the density corresponding to lipid-N may correspond to DHEA (*Figure 1—figure supplement 4E*), it is also possible that this density corresponds to a larger molecule that is only partially ordered.

## Cx43 junction interface

As in other connexins, the Cx43 GJC interface is formed by two extracellular loops, ECL1 and ECL2 (*Figure 3A, B*). The portion of the ECL1 comprising residues N55-Q58 makes contacts with two Cx43 monomers from the opposite membrane, with N55-Q58 of one monomer and Q58'-P59' of the neighbouring monomer within 4 Å distance (*Figure 3C*, *top left*). The ECL2 region that directly participates in sealing the junction includes residues P193-D197. Comparisons of the junction-forming loops in Cx43 to those in Cx46 (*Figure 2C*, *middle*) and Cx26 (*Figure 2C*, *right*) show that each protein uses

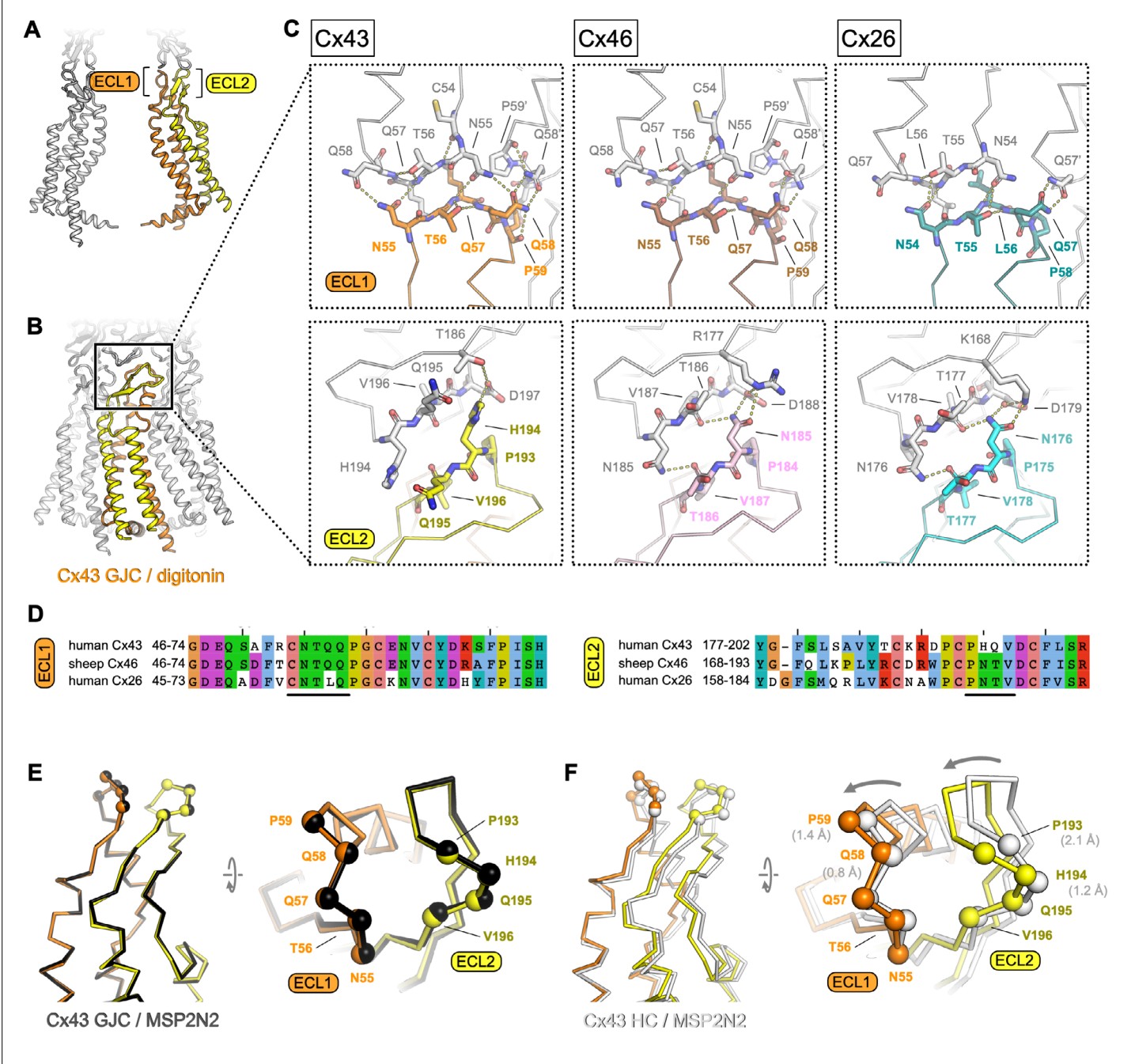

**Figure 3.** Extracellular domain of connexin-43 (Cx43) gap junction channel (GJC) and hemichannel (HC). (**A, B**) The lateral views of Cx43 GJC in digitonin, indicating the positions of the extracellular loops 1 and 2. (**C**) Views of the extracellular loops ECL1 (top) and ECL2 (bottom), for Cx43 (left), Cx46, (middle; PDB ID: 7jkc), and Cx26 (right; PDB ID: 2zw3). The residues of one monomer in each of the structures, directly involved in junction formation, are coloured by element with carbon atoms as orange/yellow (Cx43), brown/pink (Cx46) and teal/cyan (Cx26). The residues of the neighbouring connexin monomers within 4 Å distance are shown coloured with white carbon atoms. Dotted lines indicate electrostatic contacts, calculated in PyMol. (**D**) Sequence alignment of the complete ECL1 and ECL2 regions of the three proteins shown in C. The black lines indicate the GJC interface residues, as shown in C. (**E**) Alignment of the Cx43 GJC structures in digitonin micelles and in nanodiscs. Cα atoms of the interface residues shown in C are represented as spheres (the ribbon and spheres of Cx43 in nanodiscs is coloured black). (**F**) Same as E, for a comparison between Cx43 GJC in digitonin and Cx43 HC in nanodiscs. The arrow indicates the movement of the two loops that accompanies GJC formation. The ribbon and spheres of Cx43 HC are light grey. The grey numbers in brackets indicate displacement of selected Cα atoms (residues P59, Q58, P193, and P194).

a similar pattern of inter-HC interaction within each GJC. It is noteworthy that both the amino acid sequences (*Figure 3D*) and the interaction networks (*Figure 3C*) are similar in the ECL1 for Cx43 and Cx46, and in the ECL2 for Cx46 and Cx26. The differences in these two loops, and especially in ECL2, underlie the inability of Cx43 (which has been grouped into a docking group 2) to engage in heterotypic interactions with group 1 connexins (such as Cx46 and Cx26) (*Bai et al., 2018*). Our structure is thus in line with the recognized consensus motif necessary for heterotypic complementarity of connexin GJCs (*Koval et al., 2014*).

## Conformation of the Cx43 HC

Although the two Cx43 GJC structures (in detergent micelles and in nanodiscs) are almost identical, the Cx43 HC shows a notable difference in the ECL1–2 conformation (*Figure 3E, F*). The resolution of the HC reconstruction does not allow us to make definitive statements about the orientations of individual side chains, but it is clear that both loops engaging in junction formation move inward upon docking of the two HCs (*Figure 3F*). This conformational change likely involves intra- and intermolecular cooperativity. The ECL1 and ECL2 are linked via several disulphide bonds, and thus the rearrangement within each molecule would require concerted movement of the whole extracellular domain (ECD). The ECD movement within one monomer is likely cooperatively coupled to the neighbouring chains within the Cx43 HC.

## Disease-linked mutations in Cx43

A number of disease-linked mutations associated with Cx43 can be mapped directly to three regions of interest (ROI) revealed by our structures: (1) the GJC intra-pore lipid-like site, (2) the ECD, and (3) the gating region (*Figure 4A*). The interior of the channel features two lipid-like densities (*Figures 2H and 4B*; *Figure 1—figure supplement 4E*, *Figure 2—figure supplement 1A*). The observed position of lipid 1 is consistent with the previously found phospholipid-like densities inside the pore of the Cx31.3 HC (*Lee et al., 2020*). Lipid 2 inserts into the pocket formed by TM1 and TM2, parallel with the NTD. Unlike Cx31.3, where elongated densities within the pore region could be interpreted as hydrophobic tails of bound phospholipids, the density in the Cx43 GJC appears consistent with that of a sterol (*Figures 2H and 4B*). The presence of a small hydrophobic small molecule in Cx43 at this site suggests a potential mechanism of lipid-based Cx43 regulation, whereby binding of a lipid could directly influence the conformation of the gating elements of the protein (such as NTD). An effect of a cholesterol analogue 7-ketocholesterol on Cx43 permeability has been observed previously (*Girão et al., 2004*), and such an effect may be mediated via the binding sites within the GJC (or HC) pore (*Figure 4B*; *Figure 2—figure supplement 1A*). The presence nearby of several residues known to be linked to ODDD when mutated (S27P, I31M (*Richardson et al., 2004*), S86Y (*Jamsheer et al., 2014*), and L90V ) points to a potential functional significance of the intra-pore lipid-binding site.

Two known mutations linked to ODDD are located in the Cx43 ECD: P59 in ECL1 (*Vasconcellos et al., 2005*) and H194 in ECL2 (*Vitiello et al., 2005*). As suggested by the Cx43 GJC structures, these residues are located in conserved regions where any substitution can be expected to cause disruption of the contacts critical for junction formation (*Figure 4C*).

Multiple mutations in Cx43 associated with ODDD are located in the N-terminus (G2V, D3N, L7V, L11I/P, Y17S, and S18P), the TM1 region (G21R, G22E, and K23T), and the TM2 region proximal to the gate (V96M/E/A, Y98C, and 102N) (*Paznekas et al., 2009*). In some cases, the mutations reduce the ability of Cx43 to form the gap junction plaques at the plasma membrane, as is the case for Y17S or G21R (as well as L90V, located at the pore lipid-binding site) (*Lai et al., 2006*). The mutations tend to have deleterious effects on the permeability of Cx43 to ions and small molecules (*Paznekas et al., 2009*). The mutations G2V, D3N, W4A, and L7V were shown to eliminate the function of Cx43 GJCs (*Shao et al., 2012*). Interestingly, a G8V amino acid substitution has been linked to palmoplantar keratoderma and congenital alopecia 1 (PPKCA1). This mutant can form functional gap junctions and has enhanced HC activity (*Wang et al., 2015*). Mapping these sites on the structure of Cx43 allows us to gain insights into the possible mechanisms that underlie the associated disorders (*Figure 4D, E*).

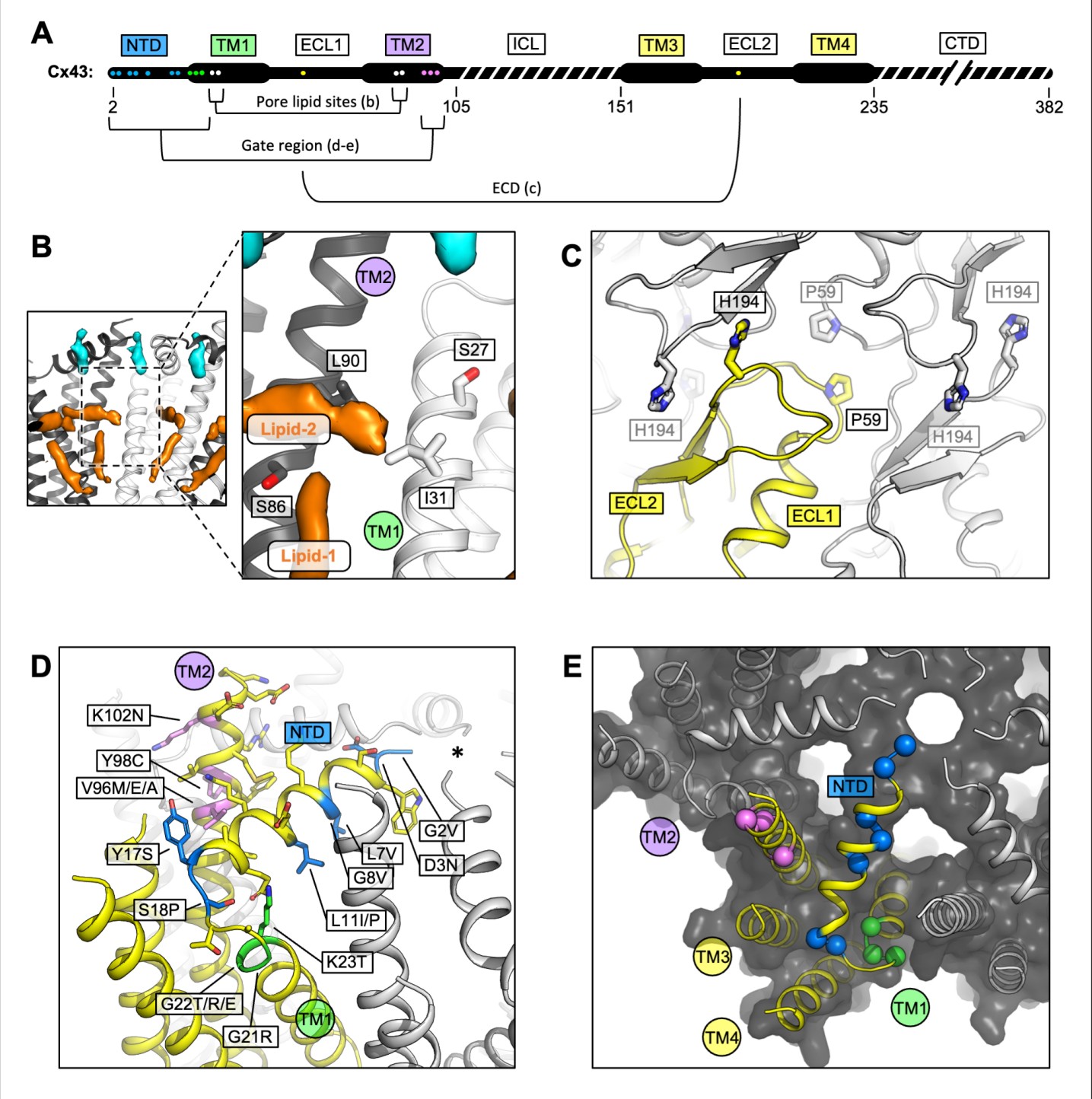

**Figure 4.** Locations of the disease-linked mutations in the connexin-43 (Cx43) gap junction channel (GJC) structure. (**A**) Disease-linked mutations mapped on the sequence of Cx43. The amino acid residues proximal to the intrapore lipid densities (Pore lipid sites) are shown as white dots. The residues at the GJC interface are shown as yellow dots (ECD). The residues of the putative gating region (Gate region) are shown as blue (NTD), green (TM1), and violet (TM2) dots. The sequence elements not resolved in our 3D reconstruction are indicated with a dashed line. (**B**) Two sterol-like density elements (lipid-1 and -2) are located within the pore region of Cx43 GJC (white). The indicated disease-linked mutations are located within close distance of the pore lipid densities. (**C**) A view of the extracellular loops forming the junction, indicating two residues known to be linked with oculodentodigital dysplasia (ODDD), P59, and H194 (side chains shown as sticks; one of the monomers in the GJC is coloured yellow). (**D**) A view of the gating region with the highlighted disease-linked mutations, coloured as in A; the asterisk indicates the center of the gating region. (**E**) A similar representation of the disease-linked mutations, with Cα atoms shown as spheres, illustrating the contribution of the mutants to gate formation.

*Figure 4 continued on next page*

*Figure 4 continued*

The online version of this article includes the following figure supplement(s) for figure 4:

**Figure supplement 1.** Molecular dynamics simulations of connexin-43 (Cx43) gap junction channel (GJC).

**Figure supplement 2.** Electrostatic properties and dimensions of the gates in connexin channels.

For example, channel blockage or hyperpermeability due to mutations in or around the gating region may link Cx43 to diseases.

## MD simulations of Cx43 GJC

Based on our model of Cx43, the dimensions of the pore opening are likely incompatible with the translocation of a larger molecule, such as ATP, cAMP, or IP3 (typical substrates of Cx43 GJC-mediated transport). To assess the permeability of the gate to ions, we performed MD simulations. We performed 18 independent MD simulations of the Cx43 GJC embedded in a double POPC bilayer solvated in 150 mM KCl. Given the ambiguity in the lipid identity and binding mode, in the first set of simulations, no potential lipid surrogates were included inside the pore or NTD region. After equilibration (detailed in Materials and methods) the dodecameric structure was stable in all simulations, as indicated by the small RMSD (*Figure 4—figure supplement 1A, D*).

The pore opening observed in our cryo-EM structures has a solvent-accessible radius of ~3 Å (*Figure 2B*). This makes it the most narrow pore opening observed for a connexin channel to date (a comparison of the pore openings in the cryo-EM structures of connexin channels is shown in *Figure 4—figure supplement 2*). However, the average solvent-accessible radius of the pore during MD was ~6 Å (*Figure 5C*); note that the effective hydrated radius of $K^+$ and $Cl^-$ is ~3.3 and ~3.6 Å, respectively. The NTD regions of Cx43 are flexible (*Figure 5—figure supplement 1*) and can move laterally and vertically (*Video 1*), allowing the passage of ions. The charge distribution inside Cx43 resembles that of the other GJCs (*Myers et al., 2018*; *Flores et al., 2020*; *Lee et al., 2020*) with a positive electrostatic potential in the NTD region of both HCs within a channel and a negative/neutral region near the HC interface (*Figure 5C*). When no transmembrane potential is applied (0 mV, *Figure 5B*), $Cl^-$ ions accumulate in the NTD regions of Cx43 GJC (anion density peaks in the NTD region) as previously observed for Cx31.3 (*Lee et al., 2020*). At 0 mV, there is an entry barrier of ~3.38 kcal/mol for cations ($K^+$) in the NTD region of both HCs (*Figure 5D*) which is slightly higher than previously reported PMFs values for other homomeric GJCs (Cx46/50) (*Myers et al., 2018*). Conversely, anions can permeate the NTD region barrierless (*Figure 5D*), but they face a higher barrier ~3.84 kcal/mol below this region where the electrostatic potential of the channel is negative, as observed for other GJC (*Myers et al., 2018*). Because the NTD domains are flexible, but they do not fully fold inwards, most ions enter and exit the same HC (*Table 2*). Application of a transjunctional voltage lowers the barrier on one HC increasing the number of permeation events. Very few full transjunctional permeation events were observed during the simulations with any applied voltage.

As mentioned before, the GJC did not fully open in any of our simulations, that is, a conformation with all NTDs within the channel symmetrically moving to open the pore was not established at any point. To illustrate this effect, we calculated the RMSD for each NTD domain in all simulations (*Figure 5—figure supplement 1*). Only at high applied transjunctional voltages only one NTD domain adopted a fully open folded inward conformation (*Figure 5—figure supplement 1*).

The MD simulations in the absence of a bound ligand reveal an intermediate, metastable state in which ions can permeate from both HCs with similar free-energy barriers and full transjunctional permeations are very rare events. The GJC selectivity and conductance properties are modulated by complex mechanisms involving both the steric aperture and the unique pattern of electrostatic features. In our case, the electrostatic properties of the channel are similar to those published for the Cx46/50 channels (*Myers et al., 2018*), but the steric contribution of the NTD region is significantly higher, increasing the barrier for $K^+$, resulting in smaller $\Delta\Delta G$ differences than those observed for the

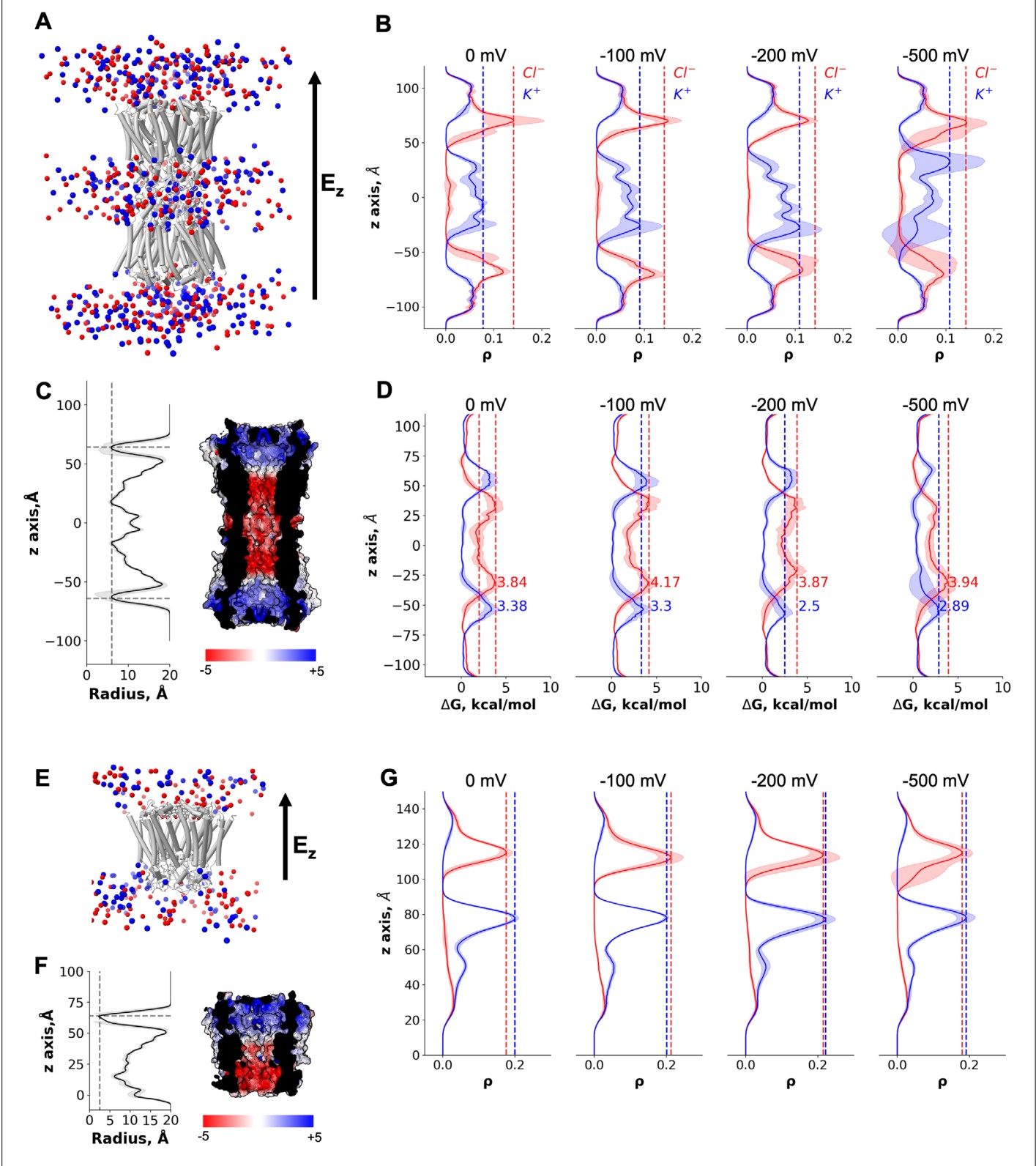

**Figure 5.** Molecular dynamics simulations of the connexin-43 (Cx43) gap junction channel (GJC) and hemichannel (HC). (**A**) Cartoon representation of the double-bilayer system including the GJC (cartoon) and ions (vdw spheres). Lipidic membrane and water residues have been removed for clarity. The direction of the applied constant electric field, $E_z$, is indicated with an arrow. (**B**) Ion density ($\rho$) profiles (average and fluctuations) along the diffusion axis of the GJC coloured in red for anions and blue for cations, for the simulated applied transjunctional voltages. (**C**) The solvent-accessible radius

*Figure 5 continued on next page*

*Figure 5 continued*

(and fluctuations during MD) along the diffusion axis of the GJC. The dotted lines correspond to the minimum radius and the position of the N-terminal domain (NTD) regions. The surface representation of Cx43 is coloured according to the calculated electrostatic potential; the slab view shows the properties of the pore. (**D**) Average free energy experienced (at least two simulation replicates per panel) by the $K^+$ (blue) and $Cl^-$ (red) while permeating the pore with different applied transmembrane potential (from right to left: 0, −100, −200, and −500 mV). The dotted lines correspond to the maximum free energy barrier for anions (red) and cations (blue). (**E**) Cartoon representation of the lipid-bound system including the HC (cartoon) and ions (vdw spheres). Lipidic membrane and water residues have been removed for clarity. The direction of the applied constant electric field, $E_z$, is indicated with an arrow. (**F**) Same as C, for the Cx43 HC. (**G**) Ion density ($\rho$) profiles (average and fluctuations) along the diffusion axis of the HC coloured in red for anions and blue for cations, for the simulated applied transmembrane voltages.

The online version of this article includes the following figure supplement(s) for figure 5:

**Figure supplement 1.** Molecular dynamics simulations of connexin-43 (Cx43) gap junction channel (GJC) at different voltages.

**Figure supplement 2.** Molecular dynamics (MD) simulations of connexin-43 (Cx43) hemichannel (HC) with dehydroepiandrosterone (DHEA) molecule in lipid-N site.

open channel (Cx46/50). Other MD studies have shown similar energy barriers for ions (***Myers et al., 2018***), leading to entry events but very rare full translocations on the MD timescale. Therefore, in the absence of the ligand the channel may exhibit a low residual conductance, as described experimentally (***Bukauskas and Verselis, 2004***).

Despite not knowing the precise identity of the lipid-N, we performed nine MD simulations of the Cx43 HC with a DHEA molecule as a lipid-N surrogate bound between the adjacent chains (***Figure 5E***, ***Figure 5—figure supplement 2A, B***, ***Video 2***), to understand whether the channel is closed in the presence of the lipid molecule. As expected, even when applying high transmembrane voltage the ion density for both species, $K^+$ and $Cl^-$, in the NTD region remained zero and no permeation events were observed (***Figure 5E-G***, ***Figure 5—figure supplement 2C***). The average pore radius during the simulations was consistent with that observed in the cryo-EM structure (***Figure 5F***). Hence, the presence of the bound lipid-N molecule seals the channel by rigidifying the NTD domains.

## Discussion

The structures of Cx43 we have obtained by cryo-EM reveal a novel conformation of a connexin channel (both GJC and HC) featuring a closed molecular gate. The state is distinct from the previously observed structures of connexin GJCs and HCs, adding an important missing component to our understanding of connexin-mediated intercellular communication. With these structures in mind, we can now propose the existence of several structurally defined gating substates of the connexin channels: (1) a fully open gate, (2) a semi-permeable central gate, and (3) a closed gate (***Figure 6***). The gating mechanisms of connexin HCs and GJC are complex and involve multiple components. The available evidence indicates that connexin channels are gated by membrane potential ($V_m$), by transjunctional potential ($V_j$), as well as by a range of chemical agents such as pH, $Ca^{2+}$, and organic molecules (***Bukauskas and Verselis, 2004***). For example, a ball-and-chain model has been proposed for Cx26, whereby at low pH the NTDs extend deep into the pore and occlude the substrate translocation pathway (***Khan et al., 2020***). This conformation is distinct from the one captured by our cryo-EM reconstructions, and it is possible that under certain conditions the Cx43 NTD may transition from the observed conformation to a low pH Cx26-like state. It is worth mentioning that for the Cx43 HCs the open probability has been shown to be very low (***Contreras***

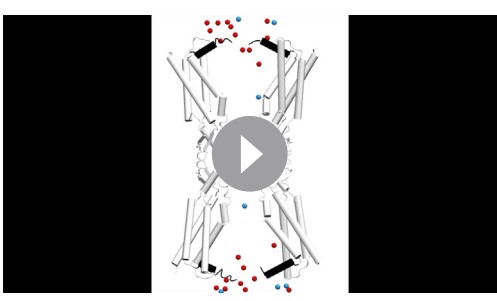

**Video 1.** Molecular dynamics simulation of the connexin-43 (Cx43) gap junction channel (GJC). N-terminal domain (NTD) of chain A opening upon ion permeation. For clarity only two chains per connexon are depicted as cartoon and coloured white. The corresponding NTDs are coloured according to their conformation position: closed (black), intermediate ('MD', red), open (blue). Ions are shown as Van der Waals spheres and coloured according to charge: positive (blue) and negative (red).

https://elifesciences.org/articles/87616/figures#video1

**Table 2.** Ion permeation event statistics.

| Applied transjunctional voltage (mV) | HC1 | | HC2 | | GJC | |
| --- | --- | --- | --- | --- | --- | --- |
| | K$^+$ | Cl$^-$ | K$^+$ | Cl$^-$ | K$^+$ | Cl$^-$ |
| 0 | 176 ± 30 | 164 ± 56 | 397 ± 119 | 686 ± 122 | 0 | 0 |
| 100 | 275 ± 114 | 126 ± 33 | 490 ± 54 | 803 ± 232 | 1 | 0 |
| 200 | 382 ± 197 | 127 ± 32 | 409 ± 101 | 926 ± 136 | 0 | 0 |
| 500 | 243 ± 149 | 196 ± 115 | 628 ± 86 | 801 ± 270 | 1 | 1 |

*et al., 2003*), and thus a closed conformation such as the one described here may represent the predominant state of the channel.

The cytosolic surface of the gate in our structures is positively charged, indicating that specific interaction with anions may be relevant to some of the gating events. As Cx43 is capable of cation and anion permeation (*Wang and Veenstra, 1997*), the presence of the positively charged surface in this region does not indicate ion selectivity. The differences in the electrostatic properties of the gates and the pore regions in different connexin channels may reflect the differences in their selectivity for ions and small molecules (*Figure 4—figure supplement 2*). However, it is also likely that the unstructured regions in Cx43 and other connexins, including the intracellular loop and the C-terminal domain, contribute to the electrostatic potential, substrate selectivity, and regulation of these channels. These domains are known to play important roles in connexin channel regulation (*D'hondt et al., 2014*). However, as these regions are unresolved in any of the available 3D reconstructions of connexin channels, the structural basis of their action remains to be determined.

The observation of lipid-like molecules bound inside the pore is intriguing. The use of detergents and lipids is currently a prerequisite for structural analysis of membrane proteins, and it is possible that under solubilization conditions the abundance of added detergent and lipid (CHS) forces the protein to interact with these molecules. Nevertheless, these interactions may correspond to the naturally occurring interactions with the endogenous lipids present in the lipid bilayer of the cell. The presence of such molecules could have important implications for HC or GJC assembly, substrate permeation and molecular gating. It remains to be determined whether Cx43 or other connexin channels harbour intra-pore lipids in the cellular membranes in vivo.

Distinct gating modes have been shown in the literature for both Cx43 HCs and GJCs, with fast transitions from an open state to a state of residual conductance, and slow transitions from an open to a fully closed state (*Bukauskas and Verselis, 2004*). The structures of Cx43 described here may represent the closed states, based on the cryo-EM- and MD simulation-based evidence. The closed state could be stabilized by a yet unknown agent that occupies the NTD lipid site formed by the arrangement of the NTDs and TM2 regions observed in our structures. It is worth noting that the majority of Cx43 channels in a GJ plaque are closed, with only a small fraction

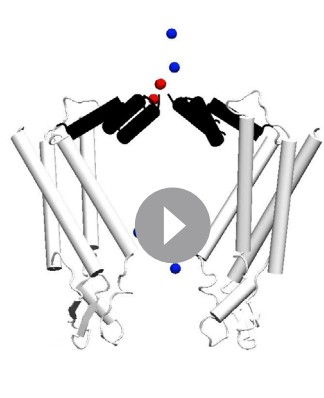

**Video 2.** Molecular dynamics simulation of the connexin-43 (Cx43) hemichannel (HC), stabilized by dehydroepiandrosterone (DHEA) molecules in the lipid-N sites. The N-terminal domain (NTD) remains closed during the whole course of the simulation (colours equivalent to those in *Video 1*).
https://elifesciences.org/articles/87616/figures#video2

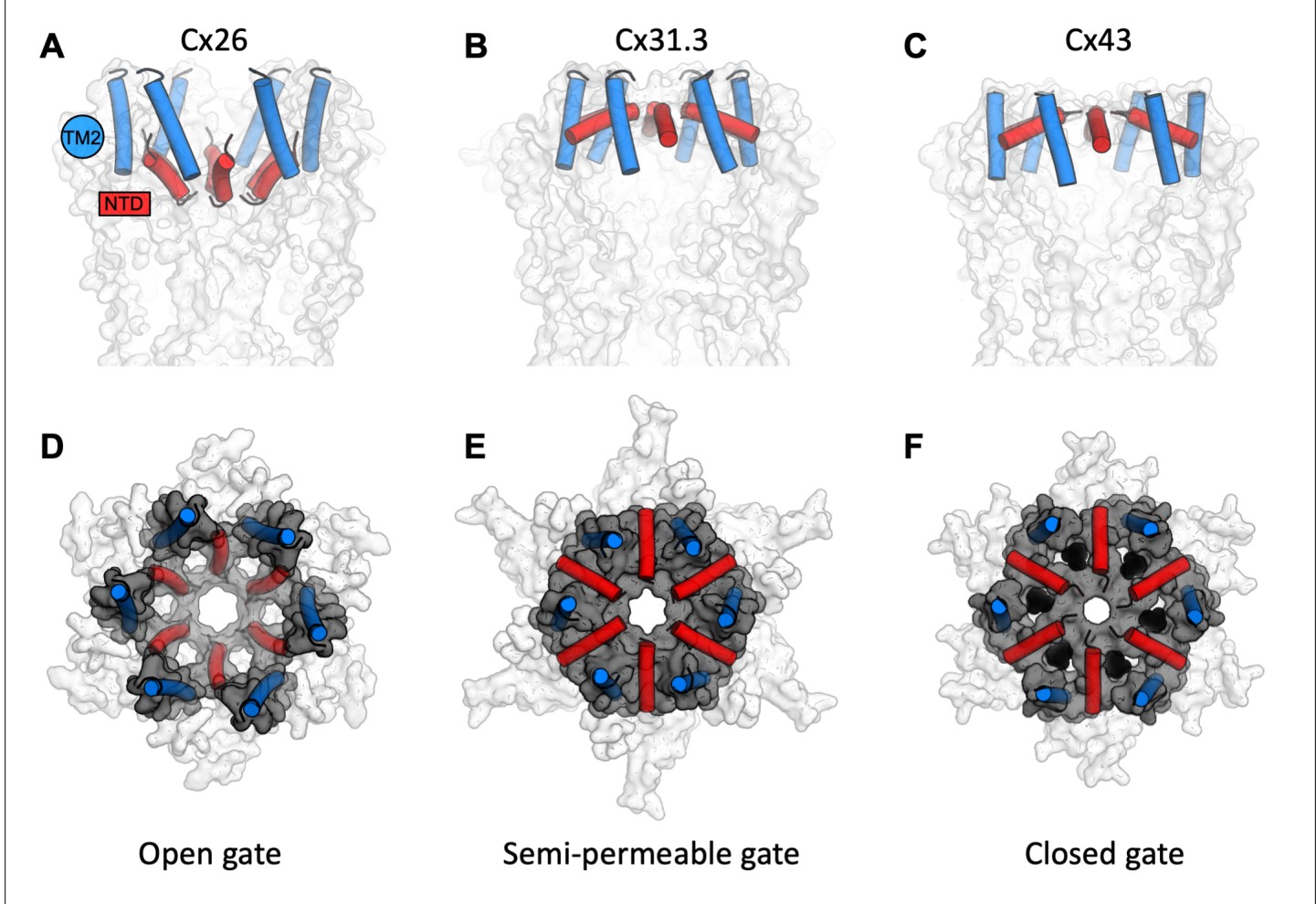

**Figure 6.** A structure-based view of connexin gating states. (**A**) The side view of a fully open gate, as observed in Cx26 gap junction channel (GJC; PDB ID: 2zw3); the gate-forming regions of the protein, N-terminal domain (NTD; red) and TM2 (blue), are shown as cylinders. Similar arrangement of the gating elements has been observed in Cx46/Cx50. This conformation of the gate is permeable to a wide range of substrates, including small molecules and ions. (**B**) The semi-permeable central gate is featured in the structure of Cx31.3 hemichannel (HC; PDB ID: 6l3t). In this conformation, the gate is likely selective for ions (Cx31.3 has been shown to have a preference for anions), based on the dimensions of the gate. (**C**) The putative closed state is featured in the connexin-43 (Cx43) structures (here, the Cx43 in digitonin is shown as an example). (**D–F**) Same as A–C, viewed from the cytosolic side of the channel. The grey surface corresponds to the NTD and TM2 regions; the white surface corresponds to the rest of the protein; lipid-N molecules in F are shown as black spheres, using a model of dehydroepiandrosterone (DHEA) manually placed into the six NTD lipid site regions for illustration purposes (the exact identity of the lipid-N molecule is unknown).

of the channels active (*Bukauskas et al., 2000*). Detailed investigations of Cx43 and other connexin channels will be required to determine whether the gate conformation observed here is a common feature among different GJCs and HCs, and to pinpoint the physiological circumstances that establish this gating conformation. Finally, capturing the structures of Cx43 channels in distinct conformations under a wide range of physiologically relevant conditions will be required to establish its mechanism of action, gating and regulation.

# Materials and methods

## Key resources table

| Reagent type (species) or resource | Designation | Source or reference | Identifiers | Additional information |
|---|---|---|---|---|
| Gene (human) | GJA1 | Uniprot | P17302 | |

*Continued on next page*

*Continued*

| Reagent type (species) or resource | Designation | Source or reference | Identifiers | Additional information |
|---|---|---|---|---|
| Cell line (human) | Freestyle 293-F cells (HEK293F cells) | Thermo Fischer Scientific; additional authentication not performed | ATCC CRL-1573 | Mycoplasma-negative (fragment length analysis performed by Microsynth) |
| Cell line (human) | HeLa DH cells | Sigma-Aldrich; aitional authentication not performed | ECACC 96112022 | Mycoplasma-negative (fragment length analysis performed by Microsynth) |
| Recombinant DNA reagent | pACMV plasmid | *Reeves et al., 2002* | N/A | |
| Recombinant DNA reagent | *Homo sapiens* Cx43 cDNA | GeneWiz | N/A | Synthetic DNA, cloned into pACMV |
| Peptide, recombinant protein | HRV3C protease | Expressed and purified by Korkhov laboratory | N/A | |
| Commercial assay or kit | NucleoSpin Plasmid | Macherey Nagel | 740588.25 | |
| Commercial assay or kit | NucleoBond Xtra Midi | Macherey Nagel | 740410.5 | |
| Commercial assay or kit | NucleoBond Xtra Maxi Plus | Macherey Nagel | 740416.1 | |
| Chemical compound, drug | Dulbecco's modified Eagle medium (DMEM) | BioConcept | 1-26F03-I | |
| Chemical compound, drug | Penicillin–streptomycin | PAN Biotech | P06-07100 | |
| Chemical compound, drug | Branched polyethyleneimine (PEI) | Sigma-Aldrich | 408727-100ML | |
| Chemical compound, drug | Trypsin–EDTA PBS 1:250 | BioConcept | 5-51F00-H | |
| Chemical compound, drug | Fetal bovine serum (FBS) | Sigma/Gibco | F0804 | |
| Chemical compound, drug | Benzamidin | Sigma-Aldrich | B6506-5g | |
| Chemical compound, drug | Leupeptin | Sigma-Aldrich | L2884-5mg | |
| Chemical compound, drug | Aprotinin | Sigma-Aldrich | A1153-10mg | |
| Chemical compound, drug | Pepstatin | Sigma-Aldrich | P5318-5mg | |
| Chemical compound, drug | Trypsin inhibitor | Sigma-Aldrich | T9003-100mg | |
| Chemical compound, drug | Phenylmethylsulfonyl fluoride (PMSF) | Sigma-Aldrich | P7626 | |
| Chemical compound, drug | Deoxyribonuclease I from bovine pancreas (DNase I) | Sigma-Aldrich | DN25-1G | |
| Chemical compound, drug | *n*-Dodecyl-β-D-maltopyranoside (DDM) | Anatrace | D310 | |
| Chemical compound, drug | Cholesteryl hemisuccinate (CHS) | Sigma-Aldrich | 1510-21-0 | |
| Chemical compound, drug | Digitonin | Merck | 11024-24-1 | |

*Continued on next page*

*Continued*

| Reagent type (species) or resource | Designation | Source or reference | Identifiers | Additional information |
|---|---|---|---|---|
| Chemical compound, drug | CNBR-activated Sepharose beads | Cytiva | GE17-0430-01 | |
| Chemical compound, drug | Superose 6, 10/300 GL | Cytiva | GE17-5172-01 | |
| Chemical compound, drug | Quantifoil R1.2/1.3 200 mesh Cupper holey carbon grids | PLANO | S143-1 | |
| Chemical compound, drug | Biobeads SM-2 adsorbents | Bio-Rad Laboratories | 152–3920 | |
| Chemical compound, drug | 1-Palmitoyl-2-oleoyl-glycero-3-phosphocholine (POPC) | Avanti Polar Lipids | 850457 C | |
| Chemical compound, drug | Tris–HCl | GERBU | 1028.25 | |
| Chemical compound, drug | Sodium chloride (NaCl) | ROTH | 3957.4 | |
| Chemical compound, drug | 4–20% Mini-PROTEAN TGX Precast Protein Gels | Bio-Rad Laboratories | 4561096DC | |
| Chemical compound, drug | Amicon Ultra-4 Centrifugal Filters Ultracel 100 K | Merck Millipore | UFC810024 | |
| Software, algorithm | Relion | Scheres, 2012. DOI: 10.1016/j.jmb.2011.11.010 | https://www2.mrc-lmb.cam.ac.uk/relion | |
| Software, algorithm | UCSF Chimera | *Pettersen et al., 2004*. DOI: 10.1002/jcc.20084 | http://www.cgl.ucsf.edu/chimera | |
| Software, algorithm | PHENIX | *Adams et al., 2010*. DOI:10.1107/S0907444909052925 | https://www.phenix-online.org | |
| Software, algorithm | COOT | *Emsley et al., 2010*. DOI: 10.1107/S0907444910007493 | http://www2.mrc-lmb.cam.ac.uk/personal/pemsley/coot/ | |
| Software, algorithm | MolProbity | *Chen et al., 2010*. DOI: 10.1107/S0907444909042073 | http://molprobity.biochem.duke.edu | |
| Software, algorithm | HOLE | *Smart et al., 1996*. DOI: 10.1016/s0263-7855(97)00,009x | http://www.holeprogram.org/ | |
| Software, algorithm | Pymol | Molecular Graphics System | https://pymol.org/2/ | |
| Software, algorithm | Prism 9.0.0 | GraphPad | https://www.graphpad.com/ | |
| Software, algorithm | GROMACS | *Abraham et al., 2015*. DOI: https://doi.org/10.1016/j.softx.2015.06.001 | https://www.gromacs.org | |

### HeLa cell culture and immunostaining

A clone of HeLa cells (DH clone, Sigma-Aldrich) that is essentially devoid of connexins was used for immunostaining and electrophysiology. Dual patch-clamp experiments (parental cells of *Figure 1—figure supplement 1*) confirmed that GJs are not present in these cells. HeLa cells were cultured in Dulbecco's modified Eagle's medium (DMEM) medium (Gibco) supplemented with 10% fetal bovine serum (FBS; Gibco) and 1% PenStrep at 37°C in a humidified incubator with 5% $CO_2$. The day after plating cells onto 12-mm diameter glass coverslips, they were transfected with a Cx43-IRES-YFP plasmid carrying the coding region of human Cx43 using Lipofectamine 3000 (Thermo Fisher Scientific).

Forty-eight hours after transfection, the cells were briefly washed with phosphate-buffered saline (PBS) and fixed using ice cold 2% PFA (paraformaldehyde) in PBS for 15 min. The cells were then permeabilized and saturated in 2% bovine serum albumin (BSA) and 0.01% Tween20 PBS solution for 1 hr at room temperature. Cells were then incubated with anti-Cx43 primary antibody (AB1727, Sigma-Aldrich, 1:100 dilution) dissolved in 1% BSA in PBS for 3 hr at room temperature. After three washes, a secondary antibody (Atto647N goat anti-rabbit IgG, 40839-1ML-F, Sigma-Aldrich, 1:250

dilution) was applied for 1 hr at room temperature. After brief washing and incubating the cells with DAPI (4',6-Diamidino-2-Phenylindole) (Merck, 1:100 dilution), the coverslips were mounted onto glass slides using Mowiol (Sigma-Aldrich) mounting medium. Fluorescence images were acquired using a Zeiss LSM 900 confocal microscope and an oil–immersion objective (63× /NA 1.4).

## Dual patch-clamp electrophysiology

Twenty-four hours after transfection, a glass coverslip with Cx43-IRES-YFP transfected HeLa cells was transferred to an experimental chamber at room temperature (22–24°C) and mounted on the stage of an upright wide-field fluorescence microscope (Olympus BX51WI) with an infinity-corrected water immersion objective (×40, 0.8 NA, Olympus). We continuously perfused cells at 2 ml/min with an extracellular solution containing 150 mM NaCl, 10 mM HEPES (4-(2-hydroxyethyl)-1-piperazineethan esulfonic acid), 5 mM KCl, 5 mM glucose, 1 mM $MgCl_2$, 2 mM $CaCl_2$, and 2 mM sodium pyruvate (pH 7.4, 311 mOsm). Cytosolic YFP fluorescence was excited by a 460-nm LED to identify the transfected cells and allow subsequent electrophysiological analysis. Patch-clamp recordings were performed using an Axon 700B amplifier (Molecular Device) with a dual headstage configuration capable of carrying out simultaneous measurements. Pipettes were filled with an intracellular solution containing 130 mM KAsp, 10 mM NaCl, 10 mM HEPES, 10 mM KCl, 1 mM $MgCl_2$, and 50 μM BAPTA (1,2-bis(o-a minophenoxy)ethane-N,N,N',N'-tetraacetic acid) (pH 7.2, 305 mOsm) filtered through a 0.22-μm pore size membrane (Millipore). DC resistance of patch pipettes in the bath ranged from 6 to 8 MOhm. Once the whole-cell configuration was achieved, both cells were voltage clamped at their common resting potential (around −20 mV). The junctional current $I_j$ was measured in cell 1 by applying voltage steps $V_j$ = +10 mV to cell 2. The corresponding junctional conductance was computed as $g_j = I_j/V_j$, without considering the potential drop $V_a$ due to pipette access resistance $R_a$, whose correction would be a source of numerical artifacts due to the large junctional currents involved. At the end of almost all experiments, we applied $CO_2$ to the bath to prove that junctional currents occurred through Cx43 GJ channels. Only cell pairs that showed complete uncoupling by the $CO_2$ were retained for the analysis. Electrophysiological data were acquired by pClamp software (version 10.4, Molecular Device) and analysed with a software we developed in Matlab (The MathWorks, Inc).

## Cx43 expression

Human Cx43 (Uniprot ID P17302) with a C-terminal 3C-EYFP-twinStrep tag, cloned into pACMV plasmid, was used for protein expression in HEK293F cells. The cells were grown in DMEM supplemented with 10% FBS and penicillin/streptomycin (PenStrep), in 15 cm plates at 37°C. Prior to transfection, the medium was exchanged to DMEM supplemented with 2% FBS and penicillin/streptomycin. The transfection mixture was prepared by mixing the expression vector and branched polyethylene-imine (PEI; Sigma-Aldrich) at a ratio of 1:2 (wt/wt). After 5-min incubation, the transfection mixture was added to the cells. After 48 hr, the cells were collected using a cell scraper, washed with PBS by centrifugation and frozen at −80°C until the day of the experiment.

## Cx43 purification

The cell pellets were resuspended in buffer A (25 mM Tris–HCl, pH 8.0, 150 mM NaCl) supplemented with protease inhibitors (1 mM benzamidine, 1 μg/ml leupeptin, 1 μg/ml aprotinin, 1 μg/ml pepstatin, 1 μg/ml trypsin inhibitor, and 1 mM phenylmethylsulfonyl fluoride), and lysed using sonication with a Vibra-Cell Sonicator (using 0.5 s pulses at 35% amplitude for 4 min). The cell membranes were collected by ultracentrifugation (Beckman Coulter Ti45 rotor, 35,000 rpm). The membranes were resuspended using buffer A with protease inhibitors and solubilized using a mixture of 1% n-dodecyl-β-D-maltopyranoside (DDM) and 0.2% cholesteryl hemisuccinate (CHS) for 1 hr at 4°C with rotation. After another round of ultracentrifugation, the supernatant was collected and mixed with 1 ml of CNBr-activated Sepharose coupled with anti-GFP nanobody (*Kubala et al., 2010*). Following a 30-min incubation at 4°C, the resin was collected and washed with 40 column volumes of buffer B (25 mM Tris–HCl, pH 8.0, 150 mM NaCl, 0.1% digitonin). The protein was eluted overnight by HRV 3C protease cleavage (200 μg) at 4°C. The eluted protein was concentrated using a 100 kDa cutoff Amicon concentrator and further purified by size exclusion chromatography (SEC) using Superose 6 Increase 10/300 GL column pre-equilibrated with buffer B. The peak fractions, corresponding to the protein of interest, were collected, concentrated, and used for cryo-EM grid preparation.

## Cx43 reconstitution in lipid nanodiscs

For nanodisc reconstitution, Cx43 protein after GFP nanobody purification was concentrated to about 300 µl. The reconstitution system was Cx43:MSP2N2:POPC = 6:4:300. POPC was first mixed with Cx43 for 30 min at room temperature, then MSP2N2 was added for further incubation at room temperature for 30 min. Bio-beads (Bio-Rad, 0.5 mg) were added to the reconstitution system. The sample was transferred to 4°C and rotated for overnight to remove the detergent. The protein was centrifuged and loaded onto Superose 6 Increase 10/300 GL column pre-equilibrated with buffer A. The peak fractions, corresponding to Cx43 nanodisc, were collected, concentrated, and used for cryo-EM sample preparation.

## Western blot

For western blot analysis, HEK293F cells were seeded in 6-well plates at a density of 500,000 cells/well in DMEM with 10% FBS and PenStrep and placed at 37°C and 5% $CO_2$ over night to allow cells to adhere. The next day the medium was exchanged to 2% FCS PSA DMEM and the cells transfected using branched PEI, with DNA to PEI at 1:2 ratio. The cells were placed at 37°C and 5% $CO_2$ for 36 hr expression.

The cells were washed twice with PBS and resuspended in 500 µl of PBS. The cells were pelleted by centrifugation at $1000 \times g$ for 15 min at 4°C and resuspended in 300 µl of 25 mM Tris–HCl pH 8.0, 150 mM NaCl, supplemented with protease inhibitors and DNase I. The cells were sonicated by a single pulse (0.5 s on, 0.5 s off) at 35% amplitude and 4× SDS–PAGE loading buffer was added to the sample. Equal volume of sample was loaded on 4–20% MiniPROTEAN TGX gradient SDS–PAGE gel (Bio-Rad) and the gels were ran at 150 V for 45 min. The transfer to nitrocellulose membrane was done in Trans-Blot SD cell (Bio-Rad) using 50 mA per gel for 1 hr in 48 mM Tris, 39 mM glycine, 20% methanol, and 0.04% SDS. The membrane was blocked in 5% BSA TBST (TBS with 0.1% Tween20) for 1 hr, and incubated with rabbit anti-Cx43 antibody (ab217676, Abcam) diluted at 1:1000 in TBST (0.1% Tween20) at 4°C over night at constant rotation. The membranes were washed three times with TBST (0.1% Tween20) and stained with goat anti-rabbit HRP-conjugated secondary antibody (ab6721, Abcam) at 1:10,000 dilution in TBST (0.1% Tween20). The membranes were washed again three times with TBST (0.1% Tween20), the signal developed using SuperSignal West Pico PLUS Chemiluminescent Substrate and imaged using Amersham Imager 600 using automatic exposure.

## Mass spectrometry-based protein characterization

### Protein digestion

A sample of purified protein (25 µg) was digested using ProtiFi S-Trap micro spin columns according to the manufacturer's protocol. After protein digestion the sample was dried in a vacuum concentrator and resuspended in 725 µl 5% acetonitrile (ACN) and 0.1% formic acid (FA). Part of the sample was transferred to an HPLC vial and further diluted 1:2 prior to analysis.

## LC–MS/MS data acquisition

1 µl of sample was analysed on an Orbitrap Eclipse Tribrid Mass Spectrometer (Thermo Fisher) equipped with a nanoelectrospray source, connected to a nano-flow LC system (Easy-nLC 1200, Thermo Fisher). Peptides were separated on a 40 cm × 0.75 µm (inner diameter) column packed in-house with 1.9 µm C18 beads at a flow rate of 300 nl/min and a 60-min linear gradient from 3% to 30% II (Eluent I: 0.1% FA, Eluent II: 95% ACN, 0.1% FA). The column was heated to 50°C. The sample was measured three times in data-independent acquisition (DIA) mode with 41 variable width DIA windows with a 1 $m/z$ overlap. For survey MS1 spectra the measured mass range was 350–1400 $m/z$ at a resolution of 120,000 with 200% normalized AGC target or 100ms maximum injection time. Survey MS1 spectra were acquired in between complete DIA isolation window sets. MS2 spectra covered a mass range of 150–2000 $m/z$ at a resolution of 30,000. HCD collision energy for MS2 spectra was set to 30% with 400% normalized AGC target or 54-ms maximum injection time.

## Peptide and protein identification

Peptide and protein identification of DIA measurements was performed using Spectronaut software (*Bruderer et al., 2015*) (Biognosys, version 14.5) in directDIA mode. Default settings were used with minor adjustments. Minimum peptide length was set to five amino acids and Phospho (STY) was

included as a variable modification. The results of the analysis are shown in *Figure 1—figure supplement 5*.

## Lipidomics analysis of purified protein

Cx43 was purified in DDM using GFP nanobody affinity chromatography with a buffer buffer 25 mM Tris–HCl, pH 8.0, 150 mM NaCl, and 0.02% DDM. Non-transfected HEK293F cells were used as a mock purification using the same purification method. The lipid was extracted by butanol–methanol single phase extraction. The extraction buffer was a mixture of $H_2O$:butanol:methanol at a volume ratio of 2:9:9. 100 µl protein sample (about 0.1 mg) was mixed with 900 µl extraction buffer and incubated at room temperature for 30 min with vortexing every 5 min. After incubation, the sample was centrifuged at 10,000 rcf/min for 10 min. The supernatant was transferred to a glass vial and used for mass spectrometry analysis.

Liquid chromatography was done as described previously (*Kind et al., 2013*) with some modifications. The lipids were separated using C18 reverse phase chromatography. Vanquish LC pump (Thermo Scientific) was used with the following mobile phases: (A) ACN:water (6:4) with 10 mM ammonium acetate and 0.1% FA and (B) isopropanol:ACN (9:1) with 10 mM ammonium acetate and 0.1% FA. The Acquity BEH column (Waters) with the dimensions 100 mm × 2.1 mm × 1.7 µm (length × internal diameter × particle diameter) was used. The following gradient was used with a flow rate of 0.6 ml/min; the following gradient was used with a flow rate of 0.6 ml/min; 0.0–2.0 min (ramp 15–30% B), 2.0–2.5 min (ramp 30–48% B), 2.5–11 min (ramp 48–82% B), 11–11.5 min (ramp 82–99% B), 11.5–12 min (isocratic 99% B), 12.0–12.1 min (ramp 99–15% B), and 12.1–15 min (isocratic 15% B).

The liquid chromatography was coupled to a hybrid quadrupole-orbitrap mass spectrometer (Q-Exactive HFx, Thermo Scientific). A full scan acquisition in positive electrospray ionization (ESI) mode was used. A full scan was used, scanning from 200 to 2000 *m/z* at a resolution of 120,000 and AGC Target 1e6 and a max injection time 200 ms, while data-dependent scans (top 10) were acquired using normalized collision energies of 20, 30, 50, a resolution of 15,000, and AGC target of 1e5. Identification of the lipids was achieved using three criteria: (1) high accuracy and resolution with an accuracy within *m/z* within 5 ppm shift form the predicted mass and a resolving power 70,000 at 200 *m/z*; (2) Isotopic pattern fitting to expected isotopic distribution; (3) the fragmentation pattern matching lipidblast library (*Kind et al., 2013*) and mzcloud (Thermo Scientific). Mass spectrometric data analysis was performed in Compound Discoverer 3.1 (Thermo Scientific) for peak picking, annotation, and matching to lipidblast.

## Cryo-EM sample preparation, data collection, and processing

The purified Cx43 protein in digitonin was concentrated to about 5 mg/ml. The Cx43 nanodisc protein was concentrated to about 1.5 mg/ml. A 3.5 µl aliquot of the protein was applied to a glow-discharged Quantifoil R1.2/1.3 200-mesh grid. The grid was blotted using Vitrobot Mark IV (Thermo Fisher Scientific) and plunge-frozen into liquid ethane. Three cryo-EM datasets were collected using a Titan Krios electron microscope equipped with a K3 direct electron detector and a GIF-Quantum energy filter (slit width of 20 eV) at ScopeM, ETH Zurich. The defocus range was set from −1.0 µm to −2.0 µm. The movies were collected in super-resolution mode using EPU 2.0, dose-fractioned in 40 frames. The exposure time for each micrograph was 0.8 s and the total dose was ~50 e⁻/Å². 

The images were assigned to distinct optics groups according to the EPU beam shift values using a script provided by Dr. Pavel Afanasyev (ETH Zurich; *Afanasyev, 2021*). The movies were binned twofold and motion corrected using MotionCor2 (*Zheng et al., 2017*), yielding aligned micrographs with a pixel size of 0.678 Å. CTF (Contrast transfer function) estimation was performed using Gctf (*Zhang, 2016*). Particle picking and 2D classification was performed in Relion 3.1 (*Zivanov et al., 2018*) (nanodisc sample) and Relion 4.0 (*Scheres, 2012*) (detergent sample). After several rounds of 2D classification, the selected particles were used for 3D classification.

For detergent sample, the best 3D classes from the 3D classifications were merged and refined to 2.61 Å. After CTF refinement and particle polishing, the resolution was improved to 2.43 Å. An additional round of 3D classification without alignment was performed, the best class was selected for 3D refinement, yielding a 2.41 Å resolution refined map. Continued refinement with a detergent-free mask yielded a reconstruction at a resolution of 2.26 Å. To further analyse the conformation viability, we performed particle symmetry expansion using *relion_particle_symmetry_expand* and subtracted

the particles with a box size of 200 pixels. The subtracted particles were further classified using 3D classification in Relion with $T$ = 20. Two 3D classes showed high-resolution features but were not substantially different from each other.

For Cx43 in nanodiscs, two datasets were processed separately for 3D classification. The best HC classes were joined together for 3D refinement to 3.56 Å. Because of the preferential orientation, two rounds of 3D classification were performed. With a detergent-free mask, the final refinement reached 3.98 Å and yielded a less anisotropic map. This map was used for model building. For GJC, the best classes from 3D classification were joined to 3D refinement to 3.25 Å. An additional round of 3D classification without alignment was performed. After CTF refinement, the resolution was improved to 2.95 Å. The pixel size was corrected during postprocessing to 0.654 Å. Local resolution maps were calculated by ResMap (*Kucukelbir et al., 2014*) implemented in Relion. Symmetry expansion and 3D classification was performed in Relion. The detailed steps of image processing are shown in *Figure 1—figure supplements 3–4*, *Figure 1—figure supplements 6–8*, and *Table 1*.

## Model building and refinement

Model building was carried out manually in Coot (*Emsley et al., 2010*). Residues L106-G150, V236-I382 were not built because these regions were not resolved in the refined density map. The M1 residue was excluded based on the MS data, which indicate that the N-terminal peptides present in the sample lack the M1 (*Figure 1—figure supplement 5*). The side chains of the bulky residues (Phe, Trp, Tyr, and Arg) were used as a guide for model building, further aided by the available connexin structures. The model was refined using real_space_refine in Phenix (*Adams et al., 2010*). The model was validated as previously described (*Qi et al., 2019*). The atoms of the final model were randomly displaced by 0.5 Å using PDB tools implemented in Phenix. The perturbed models were refined in Phenix against the half map 1. The refined model was used to generate the Fourier shell correlation curve of the model versus half map 2. The geometry of the model was validated using MolProbity *Chen et al., 2010*. All figures were prepared in PyMol (*Pettersen et al., 2004*), Chimera (*Pettersen et al., 2021*), or Chimera X (*Lee et al., 2019*).

## MD simulations

The CHARMM-GUI (*Hess et al., 2008*) server was used to build the initial simulation box comprising the full dodecameric gap junction Cx43 with one HC embedded in a POPC bilayer and fully solvated using TIP3P water. Sidechains were protonated according to neutral conditions. Disulphide bonds, identified in the experimental structure, were enforced. Amino acids corresponding to the intracellular loop (residues 106–151) and CTD (C-terminal domain) (residues 236–382) were not modelled, as the experimental data describing these large disordered portions of the structure were inconclusive, and successful MD simulations have been carried out for other dodecameric gap junctions without including these structural features (*Myers et al., 2018*; *Flores et al., 2020*). The second bilayer was added, replicating by symmetry the one embedded with CHARMM-GUI, the water placed inside the channel was kept and the rest of the system was solvated again using GROMACS (*Huang et al., 2017*) with 150 mM KCl. The final system was cleaned to remove unfavourable water positions. The topology was constructed using CHARMM36M (*Jorgensen et al., 1983*) for the protein, POPC lipids, and TIP3P (*Abraham et al., 2015*) water. The final system had 461,767 atoms: 37,320 protein atoms, 136,144 POPC atoms, 287,373 TIP3P atoms, 447 K$^+$ and 483 Cl$^-$ ions. The net charge of the gap junction is 36e$^-$. We also built a system with counterions only.

The system was equilibrated using GROMACS2021 (*Huang et al., 2017*; *Bussi et al., 2007*) starting from an initial minimization step with the protein and lipid molecules harmonically constrained during 5000 steps, followed by a 125 ps (dt = 1 fs) heating step at 303.15 K in the NVT ensemble. A second heating step of 125 ps (dt = 1 fs) was performed at 303.15 K in the NVT ensemble decreasing the lipid restraints. In the third step we decreased the lipid restraints and applied a Berendsen semi-isotropic barostat (1 atm) during 125 ps (dt = 1 fs) keeping the temperature at 303.15 K. In the fourth step (500 ps) we kept decreasing the restraints and increased the integration timestep to 2 fs, keeping a semi-isotropic barostat (1 atm) and thermostat at 303.15 K. We performed a final NPT step (30 ns) keeping the restraints on the C$_\alpha$ of the interface only. Production runs were performed with V-rescale thermostat (*Gumbart et al., 2012*) at (303.15 K) with periodic boundary conditions and particle mesh

ewald. Eighteen independent simulations (100 ns) of the GJC in the absence of ligands were started from the equilibrated structure.

The system with the DHEA molecule as a lipid-N surrogate modelled in the NTD lipid sites was equilibrated following the same protocol but the centre of mass of the DHEA molecule was harmonically restrained during equilibration and production runs, same as the NTD, with a harmonic force constants of 1000 kJ mol$^{-1}$ nm$^{-2}$.

We performed constant electric field simulations (*Tribello et al., 2014*) with different applied transjunctional voltages (−100, −200, and −500 mV) where the applied electric field was calculated as $E_z = V_{applied}/L_z$ ($L_z$ is the simulation box length along the channel diffusion axis, $z$).

RMSD calculations were calculated using GROMACS2021. Interface distances were computed using PLUMED2.7 (*Goddard et al., 2018*). All MD plots were made with the seaborn library for python. The cartoon representations included in the panels of some figures and movies were created using ChimeraX (*Schindelin et al., 2012*). For every trajectory, we computed the density of ions within a cylinder of radius 10 Å using an in-house code that provides the average (in time) number of ions in a 3D grid, spaced every 1 Å. For all grids, we computed the integral along the Cx43 diffusion axis ($z$, axis). We then computed the K$^+$/Cl$^-$ free energy profiles (*Figure 5*) along the diffusion axis of the Cx43 channel as $\Delta G = -RT \ln(\rho^-)$ and its error as $\sigma\_\Delta G = |RT \, \sigma\_\rho/\rho|$.

Using the same code (described above for the ion density calculations) for the protein-only production trajectory, we computed the solvent-accessible area and fluctuations by counting the empty grid points.

## Cell culture for light microscopy experiments

HEK293F cells were seeded at a density of 30,000 cells per well in poly-L-lysine pre-coated ibidiTreat µ-slide microscopy chambers for protein membrane colocalization studies in Dulbecco's modified Eagle medium (DMEM) supplemented with 10% FBS and PenStrep. The cells were left over night at 37°C and 5% CO$_2$ and transfected the next day. They were transfected using branched PEI with DNA to PEI in 1:2 ratio (wt/wt). For plasma membrane colocalization studies, 0.46 µg of pAYST-Cx43 (Cx43 with a YFP tag) was used. DNA and PEI dilutions were prepared separately in DMEM medium, mixed and incubated for 5 min before added to the wells or dishes in a drop-wise manner. The cells were placed in the incubator at 37°C and 5% CO$_2$ for 48 hr.

## Protein membrane colocalization studies

The cells expressing Cx43-YFP were stained with Hoechst 33342 (50 µg/ml) and Vybrant CM-DiI (1:100 dilution) in 1× PBS for 15 min at 37°C and 5% CO$_2$. They were washed two times with 1× PBS and imaged in FluoroBright DMEM supplemented with 1% FBS and PenStrep.

## Confocal light microscopy

For all purposes, the cells were imaged using Leica Stellaris 5 confocal microscope equipped with a HyD detector, using LAS X (4.2.1.23819 – build 23180) acquisition software, frame sequential data acquisition scheme using 405, 488, and 561 nm laser lines for Hoechst 33342, EYFP, and CM-DiI, respectively. The images were collected with ×20 air objective (NA 0.75) and a pinhole size of 38.3 µm.

## Confocal light microscopy image analysis

Co-localization studies were performed in Fiji (*Manders et al., 1993*) using the Coloc2 plug-in after background subtraction and 8-bit conversion of images. To confirm that the observed lines between two cells are Cx43-YFP GJCs located on the plasma membrane, the threshold Manders' co-localization coefficient (*Jurrus et al., 2018*) of YFP with CM-DiI was determined at those lines, defined as the ROI ($n = 9$; mean ± standard deviation [SD] = 0.57 ± 0.18). These coefficients were compared to coefficients of YFP in randomly selected cellular ROIs ($n = 9$; mean ± SD = 0.05 ± 0.09). The statistical significance was determined using unpaired $t$-test (p-value <0.0001, ****).

## Acknowledgements

We thank Emiliya Poghossian (EM Facility, PSI) and Miroslav Peterek (ScopeM, ETH Zurich) for expert support in cryo-EM data collection. We also thank Spencer Bliven and Marc Caubet-Serrabou (PSI) for the support in high-performance computing. The work was supported by a grant from Horten Foundation, and by the Swiss National Science Foundation grant 184951 (VMK). SAG acknowledges support from an AGAUR Beatriu de Pinós MSCA-COFUND Fellowship (project 2020-BP-00177). FLG and SAG thank the CSCS and PRACE for supercomputing resources (projects pr126 and s1107).

## Additional information

### Funding

| Funder | Grant reference number | Author |
| --- | --- | --- |
| Swiss National Science Foundation | 184951 | Volodymyr M Korkhov |
| AGAUR Beatriu de Pinos MSCA-COFUND Fellowship | 2020-BP-00177 | Silvia Acosta Gutierrez |

The funders had no role in study design, data collection, and interpretation, or the decision to submit the work for publication.

### Author contributions

Chao Qi, Silvia Acosta Gutierrez, Data curation, Formal analysis, Validation, Investigation, Visualization, Methodology, Writing – original draft, Writing – review and editing; Pia Lavriha, Data curation, Formal analysis, Investigation, Methodology, Writing – original draft, Writing – review and editing; Alaa Othman, Data curation, Investigation; Diego Lopez-Pigozzi, Formal analysis, Investigation; Erva Bayraktar, Formal analysis, Investigation, Visualization; Dina Schuster, Data curation, Investigation, Methodology, Writing – original draft; Paola Picotti, Resources, Supervision; Nicola Zamboni, Resources; Mario Bortolozzi, Conceptualization, Formal analysis, Visualization, Writing – original draft, Writing – review and editing; Francesco Luigi Gervasio, Conceptualization, Formal analysis, Funding acquisition, Investigation, Writing – original draft, Project administration, Writing – review and editing; Volodymyr M Korkhov, Conceptualization, Resources, Data curation, Supervision, Funding acquisition, Validation, Investigation, Visualization, Methodology, Writing – original draft, Project administration, Writing – review and editing

### Author ORCIDs

Chao Qi ⓘ http://orcid.org/0000-0003-1277-0363
Diego Lopez-Pigozzi ⓘ http://orcid.org/0000-0002-6178-5951
Erva Bayraktar ⓘ http://orcid.org/0000-0002-9660-5535
Dina Schuster ⓘ https://orcid.org/0000-0001-6611-8237
Mario Bortolozzi ⓘ http://orcid.org/0000-0001-7198-9838
Francesco Luigi Gervasio ⓘ http://orcid.org/0000-0003-4831-5039
Volodymyr M Korkhov ⓘ http://orcid.org/0000-0002-0962-9433

Reviewer #1 (Public Review): https://doi.org/10.7554/eLife.87616.3.sa1
Reviewer #2 (Public Review): https://doi.org/10.7554/eLife.87616.3.sa2
Author Response: https://doi.org/10.7554/eLife.87616.3.sa3

## Additional files

### Supplementary files
• MDAR checklist

## Data availability

The atomic coordinates and structure factors have been deposited in the Protein Data Bank (7Z1T, 7Z22, 7Z23); the density maps have been deposited in the Electron Microscopy Data Bank (EMD-14452, EMD-14455, EMD-14456). The mass spectrometry data have been deposited at Proteom-eXChange via PRIDE (PXD033824). The MD trajectories have been uploaded to Zenodo (8191584, 8192013, 8193133). All other data are available in the main text or the supplementary materials.

The following datasets were generated:

| Author(s) | Year | Dataset title | Dataset URL | Database and Identifier |
|---|---|---|---|---|
| Qi C, Korkhov VM | 2023 | Connexin43 gap junction channel structure in digitonin | https://www.rcsb.org/structure/7Z1T | RCSB Protein Data Bank, 7Z1T |
| Qi C, Korkhov VM | 2023 | Connexin43 gap junction channel structure in nanodisc | https://www.rcsb.org/structure/7Z22 | RCSB Protein Data Bank, 7Z22 |
| Qi C, Korkhov VM | 2023 | Connexin43 hemi channel in nanodisc | https://www.rcsb.org/structure/7Z23 | RCSB Protein Data Bank, 7Z23 |
| Qi C, Korkhov VM | 2023 | Connexin43 gap junction channel structure in digitonin | https://www.ebi.ac.uk/emdb/EMD-14452 | Electron Microscopy Data Bank, EMD-14452 |
| Qi C, Korkhov VM | 2023 | Connexin43 gap junction channel structure in nanodisc | https://www.ebi.ac.uk/emdb/EMD-14455 | Electron Microscopy Data Bank, EMD-14455 |
| Qi C, Korkhov VM | 2023 | Connexin43 hemi channel in nanodisc | https://www.ebi.ac.uk/emdb/EMD-14456 | Electron Microscopy Data Bank, EMD-14456 |
| Korkhov V | 2023 | Cx43 purified protein analysis (N-term analysis) | https://www.ebi.ac.uk/pride/archive/projects/PXD033824 | PRIDE, PXD033824 |
| Acosta-Gutierrez S, Gervasio F | 2023 | Molecular dynamics simulation data 1: Structure of the connexin-43 gap junction channel in a putative closed state | https://doi.org/10.5281/zenodo.8191584 | Zenodo, 10.5281/zenodo.8191584 |
| Acosta-Gutierrez S, Gervasio F | 2023 | Molecular dynamics simulation data 2: Structure of the connexin-43 gap junction channel in a putative closed state | https://doi.org/10.5281/zenodo.8192013 | Zenodo, 10.5281/zenodo.8192013 |
| Acosta-Gutierrez S, Gervasio F | 2023 | Molecular dynamics simulation data 3: Structure of the connexin-43 gap junction channel in a putative closed state | https://doi.org/10.5281/zenodo.8193133 | Zenodo, 10.5281/zenodo.8193133 |

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
