## [Editor Report · eLife assessment]

Gap junctions, formed from connexins, are important in cell communication, allowing ions and small molecules to move directly between cells. By determining the Cryo EM structure of the structure of connexin 43 in a putative closed state involving lipids, the study makes an **important** contribution to the development of a mechanistic model for connexin activation. The connexin 43 structure is **solid** and its presentation will appeal to the channel and membrane protein communities.

---

## [Referee Report · Reviewer #1 (Public Review)]

Gap junctions, formed from connexins, are important in cell communication, allowing ions and small molecules to move directly between cells. While structures of connexins have previously been reported, the structure of Connexin 43, which is the most widely expressed connexin and is important in many physiological processes was not known. Qi et al used cryo-EM to solve the structure of Connexin 43. They then compared this structure to structures of other connexins. Connexin gap junctions are built from two "hemichannels" consisting of hexamers of connexins. Hemichannels from two opposing cells dock together to form a complete channel that allows the movement of molecules between cells. N-terminal helices from each of the 6 subunits of each hemichannel allow control of whether the channels are open or closed. Previously solved structures of Cx26 and Cx46/50 have the N-termini pointing down into the pore of the protein leaving a central pore and so these channels have been considered to be open. The structure that Qi et al observed has the N-termini in a more raised position with a narrower pore through the centre. This led them to speculate whether this was the "closed" form of the protein. They also noted that, if only the protein was considered, there were gaps between the N-terminal helices, but these gaps were filled with lipid-like molecules. They, therefore, speculated that lipids were important in the closure mechanism. To address whether their structure was open or closed with respect to ions they carried out molecular dynamics studies, and demonstrated that under the conditions of the molecular dynamics ions did not traverse the channel when the lipids were present.

Strengths:

The high resolution cryo-EM density maps clearly show the structure of the protein with the N-termini in a lateral position and lipid density blocking the gaps between the neighbouring helices. The conformation that they observe when they have solved the structure from protein in detergent is also seen when they reconstitute the protein into nanodiscs, which is ostensibly a more membrane-like environment. They, therefore, would appear to have trapped the protein in a stable conformational state.

The molecular dynamics simulations are consistent with the channel being closed when the lipid is present and raises the possibility of lipids being involved in regulation.

A comparison of this structure with other structures of connexin channels and hemichannels gives another representation of how the N-terminal helix of connexins can variously be involved in the regulation of channel opening.

Weaknesses:

While the authors have trapped a relatively stable state of the protein and shown that, under the conditions of their molecular dynamics simulations, ions do not pass through, it is harder to understand whether this is physiologically relevant. Determining this would be beyond the scope of the article. To my knowledge there is no direct evidence that lipids are involved in regulation of connexins in this way, but this is also an interesting area for future exploration. It is also possible that lipids were trapped in the pore during the solubilisation process making it non-physiological. The authors acknowledge this and they describe the structure as a "putative" closed state.

The positions of the mutations in disease shown in Figure 4 is interesting. However, the authors don't discuss/speculate how any of these mutations could affect the binding of the lipids or the conformational state of the protein.

It should also be noted that a structure of the same protein has recently been published. This shows a very similar conformation of the N-termini with lipids bound in the same way, despite solubilising in a different detergent.

---

## [Referee Report · Reviewer #2 (Public Review)]

The authors have addressed most of the concerns. Yet, I still think the authors should at least mention in the article the residues involved in the intra-pore lipid binding pockets for further experimental validation (not only for those residues involve in disease). This is important because the lipid-like density information usually does not come integrated into the PDB structures, so it is not easily accessible for non-structural biologists. The structural data seems solid, and the MD data supports the notion that the GJC is in a putative close state.

---

## [Author Response]

The following is the authors’ response to the original reviews.

Major Revisions:1. Although we appreciate this work was carried out independently, it would improve this paper if this structure presented here was compared to the recently published structure of Cx43 (Nat Commun 14, 931 (2023)) with the conclusions including added in the discussion.

We encourage the readers to read both our study on Cx43 and the one mentioned by the reviewer. However, we believe the optimal format for such a comparison is going to be a more comprehensive review article, which is outside the scope of our study.

1. Please elaborate on the lipid-binding pockets observed for lipid 1, lipid 2, and the N-lipid/PGL. For example, what are the residues involved in these lipid-protein interactions? Are these residues conserved in other connexin isoforms? Do these lipid-binding pockets match with previous structures, including the recent Cx43 structure? Please clarify what lipid sites are ambiguous due to insufficient resolution.

Within the scope of our study, we have shown that some of the disease-linked residues are located in close proximity to the lipid sites (Fig. 4b). This suggests a possible role of the lipid sites in diseases associated with Cx43 mutations (and possibly with the mutations in other connexins, as the structures of other connexin channels also feature bound lipids inside the pore region). We feel that a more in-depth comparison will require a careful study, beyond the analysis that we have performed here, and for this reason we would like to reserve such a detailed comparison for our future work (possibly a comprehensive review article on connexin structure and function).

1. The NT domain and TM2 segments are referred to as the gate region. If there is no strong evidence to support this claim then please use "putative" gate region.

We have updated the text accordingly, referring to this region as a putative gate region where appropriate.

1. It is mentioned that there is a reorientation of extracellular loops 1 and 2 after Gap junction formation. Based on their structures, I wonder how this rearrangement alters the channel conduction pathway. For example, Do the electrostatic surface and hydrophobic properties change? Please consider adding further details as this information could be useful to understand why some properties of hemichannels differ from intercellular GJ channels.

We have updated the Fig. 5 with an illustration of the Cx43 HC surface coloured according to electrostatic potential (to match the same representation of the Cx43 GJC). It is obvious that the rearrangement of the extracellular loops 1 and 2 do not dramatically alter the electrostatic properties of the HC relative to the GJC. A more obvious difference is in the local environment of the ECLs: it is radically different in a “free” HC (exposed to the solvent or to the extracellular space of a cell), compared to the ECL environment in a connexon within a GJC (which is sealed by a docked connexon from the opposite membrane).

1. Related to the previous point, the pore profile shown in Figure 5C shows that there is a constriction site in the extracellular part with the same diameter as the observed constriction caused by the NT domain. This constriction point seems to be associated with the high energies calculated for Cl-. Please clarify if this constriction is produced by the formation of the GJC or is also present in HC?

This is the same constriction zone, and the Cl- barriers are further down the channel axis where the electrostatic potential of the protein is negative. We have included a similar calculation for the HC simulation in Fig. 5 (revised Fig. 5f).

1. Related to the MD simulations shown in Figure 5d: if the voltage is applied across the whole GJC, the free energy under voltage should not be symmetric. Please clarify.

The symmetry observed in the free energies is due to the fact that the ions enter and exit from the same hemichannel. Only at very high voltages we observe some rare full GJC permeation events, slightly unbalancing the free energy at 500 mV.

1. The scheme in Figure 6 many needs further editing. The authors propose a putative closed state in which lipids are bound next to the NT, but we suggest it should be made clearer in the figure that this is a putative model, since there is no functional evidence supporting the role of these lipids in the gating/permeation properties of Cx43. Also, please clarify what is meant by a "semi-permeable gate" - a channel that only permeates ions but not molecules?

We have updated the legend of the figure 6, to clearly reflect that this is a putative model. The “semi-permeable” state of the channel is something that was suggested previously by the authors of the Cx31.3 study, and we refer to that structure in the figure.

Minor comments:1. In the result section there are some statements that currently lack solid experimental support. Please consider editing or moving this text to the discussion section only. A good example of this is the Diseaselinked mutation section, specifically lines 199-206. In another example: in lines, 237-238 authors state that NT can move laterally and vertically, but this idea still requires experimental validation.

We feel that the original formulations of these portions of the text are appropriate. Disrupting them would interrupt the flow of the manuscript, and we prefer to stay with the original text in this case.

1. Line 283. "With these structures in mind, we can now establish the existence of several structurally defined gating substates of the connexin channels". Please, tone down this statement. Replace "establish" with "propose" or another more appropriate word.

We have updated the text as suggested ("propose” instead of “establish”)

1. Line 313-314. " The presence of such molecules could have important implications for HC or GJC assembly, substrate permeation, and molecular gating". Currently, this entire statement does not have any support. Is there any paper that authors can discuss to suggest with some basis that lipids might have a role in assembly, permeation or gating?

We feel that this statement is sufficiently careful, conveying a thought that the presence of such molecules could have important implications for various HC- or GJC-related processes. It is not a particularly strong claim and seems to be appropriate in this context.

1. It seems that the structure shown in panels A and C in Figure 2 are shown in opposite directions, which makes the figure confusing. If needed, please rotate the structure in panel A to show the cytosolic part of the protein as panel C. Also, in the same figure, panels G and F are wrongly labeled. Please correct.

For Fig. 2a, the angle is very different from anything else we show in the figure, so we would rather keep this as it is now. We have corrected the labelling for Fig. 2g-h.

1. Check spelling mistakes in the legend of Extended data Fig.2, Extended data Fig.9, and line 243.

We are grateful to the reviewers for pointing out the typos, which have now been corrected.

1. The colors for G-L isoforms are not specified in Extended Data Fig.10. Please correct this.

We updated the figure, removing the PGL label (the correct label is “lipid-N”).

1. It is not clear what is the difference between PGL and the N-lipid density. Does PGL refers to the lipid-like density observed in nanodiscs, as indicated in Extended Fig. 4 and 10?. Please clarify this issue in the manuscript.

The labeling has been corrected in like with the revised version of the manuscript (this density element is now referred to as the “lipid-N”).

1. Page 7 line 234-235 "The pore opening has a solvent-accessible radius of ~6Å (Figure 5c) very close to the effective hydrated radius of K+ (~6.6 Å) and Cl- (~7.2 Å). This makes it the most narrow pore opening...", it should be diameter, not radius.

We have added a calculation for the HC (new Fig. 5f) and corrected the text as follows (line 234):

“The pore opening observed in our cryo-EM structures has a solvent-accessible radius of ~3 Å (Figure 2b). This makes it the most narrow pore opening observed for a connexin channel to date (a comparison of the pore openings in the cryo-EM structures of connexin channels is shown in Extended Data Fig. 12). However, the average solvent-accessible radius of the pore during molecular dynamics was ~6 Å (Figure 5c); note that the effective hydrated radius of K+ and Cl- is ~3.3 Å and ~3.6 Å, respectively.”

And line 277:

“The average pore radius during the simulations was consistent with that observed in the cryo-EM structure (Fig. 5f).”